# Discharge domains regulation and dynamic processes of direct-current triboelectric nanogenerator

Jiayue Zhang [1,4], Yikui Gao[2,3,4], Di Liu [2,3,4], Jing-Shan Zhao [1]✉ & Jie Wang [2,3]✉

Direct-current triboelectric nanogenerators arising from electrostatic breakdown can eliminate the bottleneck problem of air breakdown in conventional triboelectric nanogenerators, offering critical benefits of constant-current output, resistance to electromagnetic interference, and high output power density. Previous understanding is that its output characteristics are described by a capacitor-breakdown model or dictated by one or two discharge domains in direct-current triboelectric nanogenerators. Here, we demonstrate that the former holds only for ideal conditions and the latter cannot fully explain the dynamic process and output performance. We systematically image, define, and regulate three discharge domains in direct-current triboelectric nanogenerators, then a "cask model" is developed to bridge the cascaded-capacitor-breakdown dynamic model in ideal conditions and real outputs. Under its guidance, the output power is increased by an order of magnitude within a wide range of resistive loads. These unexplored discharge domains and optimization methods revolutionize the output performance and potential applications of direct-current triboelectric nanogenerators.

With the rapid development of the fifth-generation (5G) networks and artificial intelligence (AI) technology, the era of the Internet of Things (IoTs) is inevitably coming with an exigent requirement for a vast sensing network and abundant energy, which is difficult to satisfy under the current energy crisis and environmental pollution conditions[1–3]. Therefore, clean, reliable, sustainable, and abundant power supplies are in great demand[4–7]. Triboelectric nanogenerator (TENG) with the merits of a broad selection of materials[8], low cost of manufacturing[9], lightweight configuration[10], strong scalability[11], the high conversion efficiency of energies at low motion frequencies[12], and without pollution[13], that can directly convert environmental energy into electricity based on contact electrification and electrostatic induction effects[14], is an eco-friendly power generation option to satisfy the great energy demand in the IoTs[15–19]. The performance of TENG, which is usually characterized by power density, is theoretically

proportional to the square of its surface charge density[20]. However, air breakdown has been demonstrated as the bottleneck problem in all working modes of TENGs, serving as a limitation on further increases in surface charge density[21–24].

Alternatively, the emergence of direct current triboelectric nanogenerator (DC-TENG)[25], which can effectively make use of air breakdown, eliminates this bottleneck problem by regulating air breakdown, clearing the way for surface charge density boosting and making it possible to greatly improve output performance[26,27]. In addition to the advantages of the conventional TENGs, DC-TENGs have the unique advantages of constant current output characteristics, resistant to electromagnetic interference, and not being limited by electrostatic breakdown of the dielectric materials, making them appropriate for directly powering electronics and benefiting the future development of the IoTs. To enhance the output performance of

[1]Department of Mechanical Engineering, Tsinghua University, Beijing 100084, P. R. China. [2]Beijing Institute of Nanoenergy and Nanosystems, Chinese Academy of Sciences, Beijing 101400, P.R. China. [3]School of Nanoscience and Technology, University of Chinese Academy of Sciences, Beijing 100049, P.R. China. [4]These authors contributed equally: Jiayue Zhang, Yikui Gao, Di Liu. ✉e-mail: jingshanzhao@mail.tsinghua.edu.cn; wangjie@binn.cas.cn

DC-TENGs, efforts have been made to strengthen air breakdown through materials selection[28], configuration optimization[29–31], and environment control[32]. Besides experimental research, a corona discharge model was proposed to explain the discharge process between the charge collecting electrode (CCE) and triboelectric layer (TL) in DC-TENGs[33], and a simulated algorithm was reported to practically calculate the output performance of DC-TENGs[34]. Undoubtedly, these efforts have greatly boosted the output performance of DC-TENG and stimulated the fast development of this research field. However, their understanding of the discharges in DC-TENGs is inadequate, and only one discharge domain-between CCE and TL-is considered for experiments. Despite another harmful breakdown domain was mentioned in a physical model, it was only a compensation in numerical solution and not observed in experiments[34]. Overall, it is commonly agreed in previous research that (i) the output performance (current, voltage, and power) of DC-TENG is decided by one discharge domain, and (ii) the theoretical model without considering other discharge domains is rather limited in practical applications. These agreements, however, make it difficult to explain the experimental observation where output charges decay at large external loads, and could hamper and even mislead the optimization direction of DC-TENG. Therefore, a comprehensive model considering all the discharge domains is required to understand the working process of DC-TENGs, guide the design and optimization of DC-TENGs, and greatly improve the output power.

In this work, a cascaded-capacitor-breakdown model was presented for the dynamic output of DC-TENGs in ideal conditions, and all the potential discharge domains in a DC-TENG were systematically imaged, defined, and regulated for output optimization. Starting with an analysis of a single capacitor breakdown model extracted from CCE and TL, an expanded dynamic model with a semi-empirical numerical algorithm was developed. With careful configuration design and experiments, the model has been verified. Through observation experiments, not only the existence of multiple electrostatic breakdowns on DC-TENG were confirmed, but their roles in outputs were intuitively demonstrated. With finite element method simulation and experimental confirmations, all the possible electrostatic breakdown domains were identified and regulated. Based on that, an optimization strategy and a specific solution were demonstrated. The experimental design methods presented in this paper can also serve as a reference for future research. The dynamic model, semi-empirical simulation method, and optimization method introduced in this paper benefit the further optimized design of DC-TENG and facilitate the understanding of electrostatic breakdowns occurring during DC-TENG operation, opening a new chapter for DC-TENG performance with ultrahigh output.

## Results
### Cascaded-capacitor-breakdown model and experimental validation under ideal conditions

Under ideal conditions, except for a small amount of residual charges left on the TL surface, charges generated by the friction between FE and TL are released by electrostatic breakdown at CCE. This is the assumption of the following model and is also commonly accepted by previous studies[27,29]. As assumed, the working process of a DC-TENG is similar to that of a hydroelectric energy generating system, as the charge transfer resembles the flow of working fluid in the system. (Fig. 1a and Supplementary Note 1). Friction between FE and the TL generates net opposite charges on both contact surfaces due to contact electrification (Fig. 1b(i)). (To avoid possible confusion and enhance the readability, the term "net charge" will be referred to as simply "charge" in the following text[35,36]. Supplementary Note 2 shows more analysis and details.) As FE slides on TL, when the charged TL segment slides below CCE, the opposite charges are induced in CCE, resulting in a strong electric field between CCE and TL. Once the electric field intensity is higher than the breakdown threshold, the

electrostatic breakdown occurs between CCE and TL[37], resulting in the negative charges transferring from TL to CCE, and there will be charges flowing into the external circuit (Fig. 1b(ii)). When the relative motion between FE and TL stops, no charge is generated and the electric field intensity between CCE and TL drops below the breakdown threshold, resulting in no charge transfer between CCE and TL and no output signals (Fig. 1b(iii)).

The charge flow transferred at CCE can be modeled as a capacitor breakdown model, which is characterized as a capacitor $C_{CCE-TL}$ (made of the two facing surfaces of CCE and TL before electrostatic breakdown), a Zener diode $D_{Z,CCE-TL}$ (judgement for electrostatic breakdown), and a resistor $R_{CCE-TL}$ (equivalent resistor between CCE and TL when the electrostatic breakdown happens), as shown in Fig. 1c. As charges continuously flow into the capacitor $C_{CCE-TL}$, the electric field intensity between CCE and TL keeps increasing until the breakdown threshold, which is 3 MV m$^{-1}$ in the air, is reached. (Discussion about the breakdown threshold is shown in Supplementary Fig. 1 and Supplementary Note 3) When the electric field intensity exceeds the threshold, the electrostatic breakdown happens and the circuit is turned on, the discharge path between CCE and TL serving as a resistor $R_{CCE-TL}$.

The charge generation is continuous when the FE continuously slides on TL, while the electric output is possibly discrete or continuous, resulting in pulsed, direct-current, or even constant-current signals, which depend on the specific structure configuration and working conditions (Supplementary Fig. 2 and Supplementary Note 4). The rate of charge flowing into $C_{CCE-TL}$ can be modeled as the product of the surface charge density collected at CCE $\sigma_{CCE}$ (transferred charge measured by an electrometer divided by the area swept over by relative sliding), the width of CCE $w_{CCE}$, and relative sliding speed $v_{sliding}$ (Supplementary Note 5). To account for practical factors that result in an inconsistent rate of charge generation, a random factor $\alpha$ is introduced in the charging process (Supplementary Fig. 3 and Supplementary Note 6). In the discharging process, $C_{CCE-TL}$ is divided into multiple secondary capacitor units for better reflecting real-world conditions (Supplementary Fig. 4 and Supplementary Note 7). With these two modifications, the cascaded-capacitor-breakdown simulation model was illustrated in Fig. 1d with a better characterization of the real-world situation.

In this ideal cascaded-capacitor-breakdown simulation model, as assumed at the beginning, $\sigma_{CCE}$ can be determined by performing preliminary experiments which measured the charge collected in CCE and dividing this value with the area swept over by relative motion. The width of CCE $w_{CCE}$ is a fixed intrinsic property of the device, which is determined once fabricated. Sliding speed $v_{sliding}$ is a variable that depends on the working conditions, which can be designed and set in experiments. Calculated from these variables, the inlet rate of charge flow is evenly divided into portions and accumulated in each secondary capacitor until the breakdown threshold is reached (Supplementary Note 8). The output in the external circuit can be calculated as the sum of the output current of each individual secondary capacitor (Supplementary Fig. 5).

To verify this cascaded-capacitor-breakdown simulation model, a series of experiments were conducted on a conventional DC-TENG. Setting $\sigma_{CCE}$ 53.91 μC m$^{-2}$ (Supplementary Fig. 6), $w_{CCE}$ 3.5 cm, $v_{sliding}$ 1, 5, 10, 25, 50, and 75 mm s$^{-1}$, the simulated results for the dynamic output of DC-TENG are shown in Fig. 2a. More details are shown in Supplementary Figs. 6–8 and Supplementary Note 8. Experiments were carried out to evaluate the results. A polyvinyl chloride (PVC) film was adhered on an acrylic substrate serves as TL. A device with $w_{CCE}$ of 3.5 cm was sliding on it under the six speeds, and the output currents were measured and shown in Fig. 2b. To better compare the simulated results and experimental data, the root-mean-square (RMS) current, which is the effective current, and the crest factor (CF), which is the ratio of peak current to the effective current (Supplementary Note 10),

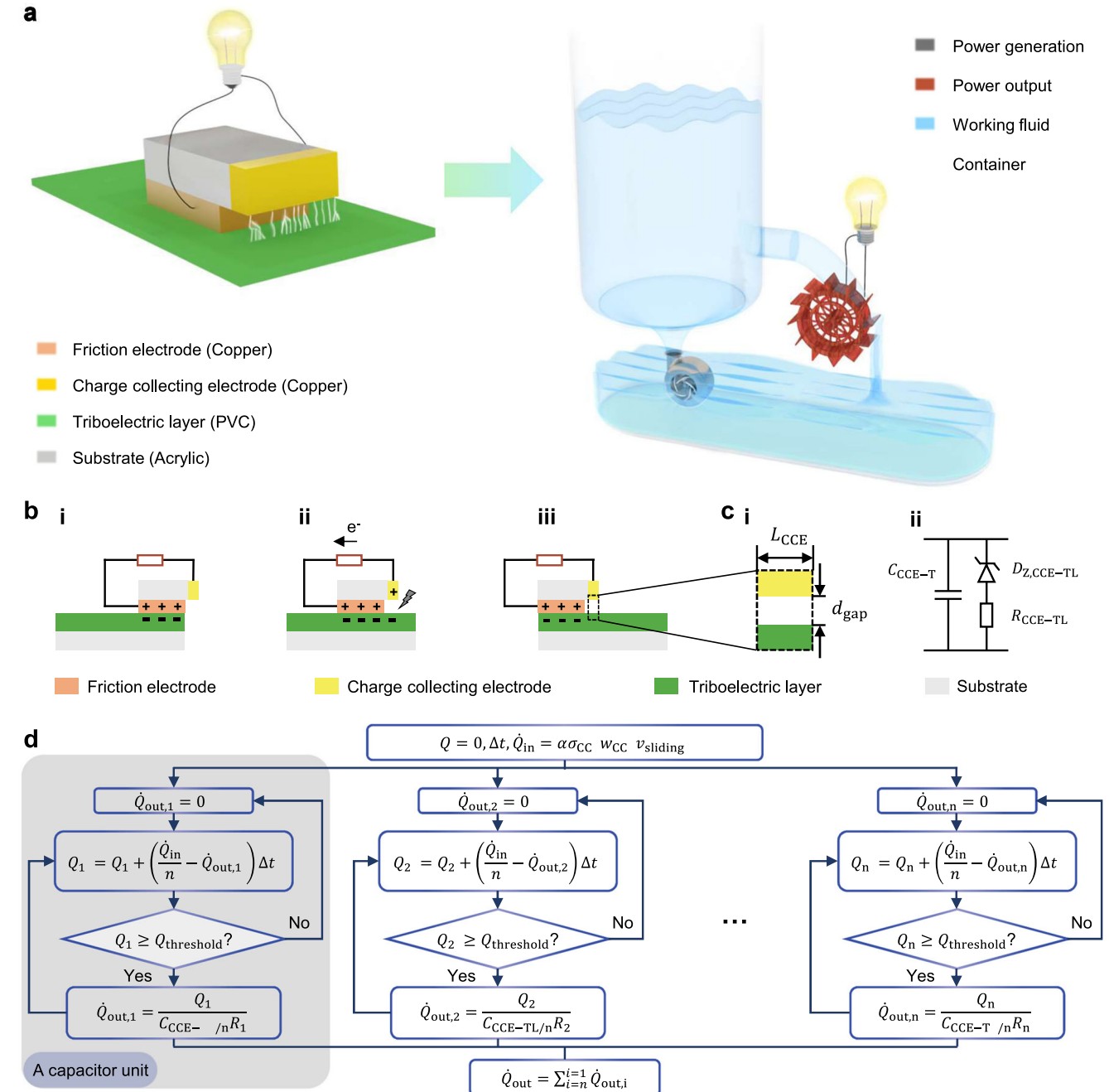

**Fig. 1 | Illustration for the cascaded-capacitor-breakdown model of DC-TENG. a** Schematic diagram of the working process of DC-TENG. **b** Working mechanism of DC-TENG. **c** Equivalent circuit for the discharge process between CCE and TL. **d** The cascaded-capacitor-breakdown model for the dynamic output of a DC-TENG.

were calculated for each group. As shown in Fig. 2c, the experimental data points fall in the 95% confidence band of the simulated results, and the CFs of the experimental data and the simulated results are also close (Fig. 2d).

Furthermore, the three main variables in the cascaded-capacitor-breakdown model were all tested. To examine the influence of $\sigma_{CCE}$, TLs made of three materials (polyether ether ketone (PEEK), PVC, and ethylene tetrafluoroethylene (ETFE)) with varying contact electrification capabilities and electrostatic breakdown capabilities were fabricated (Supplementary Note 11). With a relative displacement of 52.5 mm, the transferred charge was presented in Fig. 3a. Output current was measured at $v_{sliding}$ of 10 mm s$^{-1}$ (Fig. 3b). Figure 3c shows RMS current linearly increasing with $\sigma_{CCE}$, which indicates better output can be achieved by selecting appropriate material-pairs (Supplementary

Note 12). To evaluate the effect of $w_{CCE}$, experiments using devices with widths of 1, 2, and 3 cm were carried out. By sliding on a PVC film with a relative displacement of 52.5 mm, the charge transferred between FE and CCE and the currents in the external circuit under a constant $v_{sliding}$ of 10 mm s$^{-1}$ are proposed in Fig. 3d–f. The increase in $w_{CCE}$ leads to a linear increase in RMS current and a decrease in CF. For the third variable, sliding speed, a device with $w_{CCE}$ of 3.5 cm was fabricated and evaluated by sliding on PVC with $v_{sliding}$ of 10, 50, and 100 mm s$^{-1}$. It is worth noting that with the same displacement, the amount of charge transferred is not strictly consistent due to a certain degree of randomness in the contact electrification process, which was already considered in the cascaded-capacitor-breakdown model by the parameter $\alpha$. Figure 3g–i present the experimental data with awesome linearity between $v_{sliding}$ and the output current.

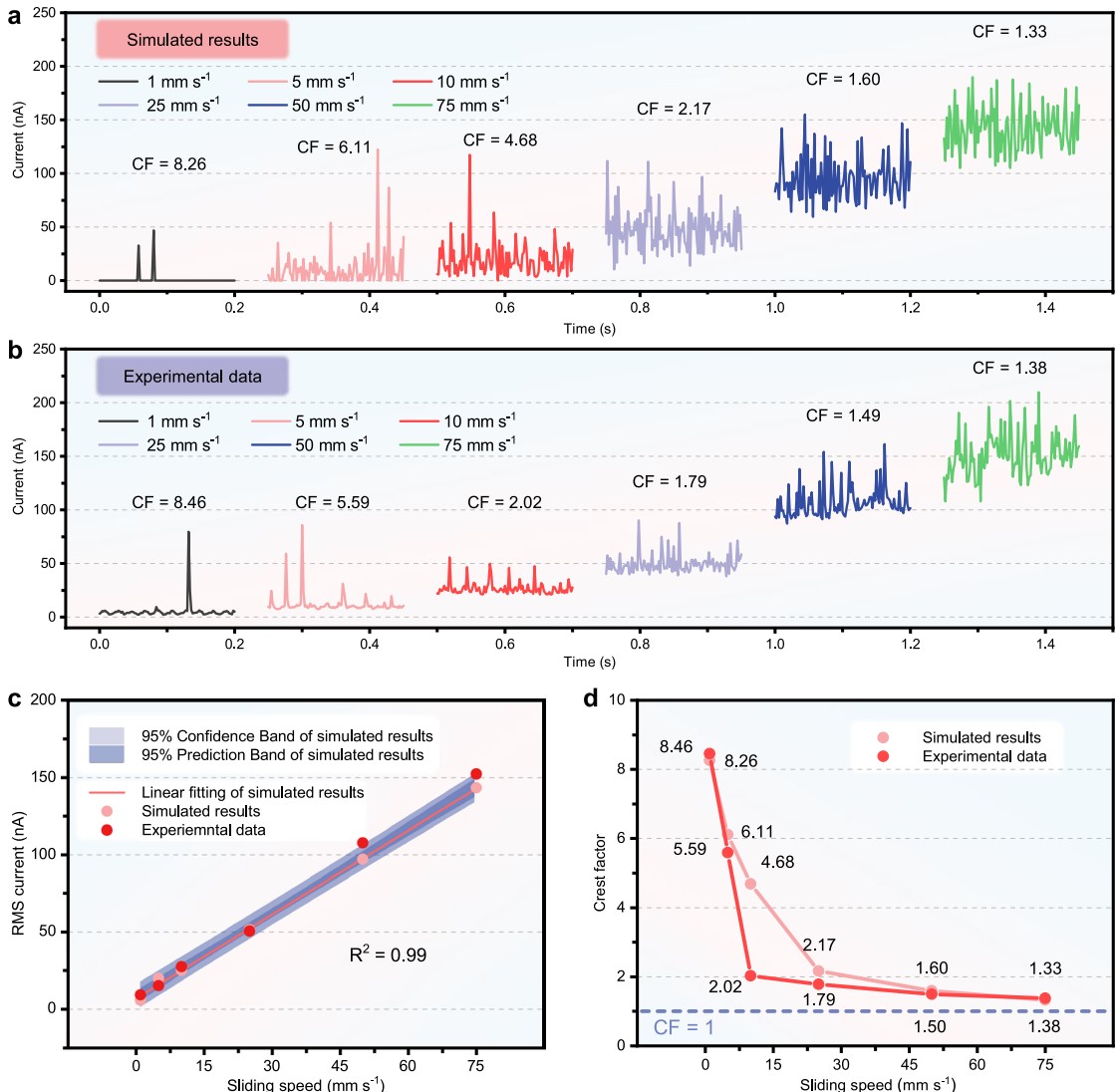

**Fig. 2 | Output current at different sliding speeds obtained in experiments and simulations. a** Output current obtained in simulations. **b** Output current obtained in experiments. **c** RMS currents calculated from the experimental data and simulated results. **d** CFs calculated from the experimental data and simulated results. Source data are provided as a Source data file.

## Identification and regulation of three discharge domains

The model previously described only holds under ideal conditions, namely the assumption that except for a small amount of charges remaining on TL surface, triboelectric charges produced at FE are electrostatically broken-down at CCE. However, this assumption fails to account for the presence that electric field intensities in multiple locations exceed the breakdown threshold in numerical solutions, as well as the non-conservation between triboelectric charges and electrostatic breakdown charges. We found that there are totally three discharge domains around the DC-TENG. Based on the characteristics of the discharges, we defined the discharge regions as the 1st domain, 2nd domain, and 3rd domain (Fig. 4a). The 1st domain is the electrostatic breakdown region between CCE and TL, which is the region discussed by the cascaded-capacitor-breakdown simulation model above and has been the focus of numerous previous research as the sole discharge domain. The electrostatic breakdown here is triggered by the high electric field intensity between a conductor (CCE) and a dielectric layer (TL). The physical mechanism behind it is not clear, but some studies have suggested that it is a kind of corona discharge, which emits weak light (Fig. 4b). The 2nd domain indicates the region where the electrostatic breakdown occurs between FE and TL. It has the same electric

properties as the 1st domain, as it is also an electrostatic breakdown between a dielectric layer and a conductor (Fig. 4c). The charges released in this domain were not measured by the electrometer in the external circuit and therefore wasted without doing work in the external circuit. The 3rd domain denotes the region where the electrostatic breakdown between FE and CCE occurs. The 3rd discharging path is formed by the electrostatic breakdown between two conductors. This type of discharge typically emits a bright and clear electric arc connecting the two electrodes under a high potential difference (Fig. 4d)[38]. In the working process of DC-TENG, this kind of discharge typically arises in instances of high resistive or capacitive loads within the external circuit, as well as when the DC-TENG is employed as a high-voltage source. Correspondingly, the electrostatic breakdown that occurs in the 1st, 2nd, and 3rd domain is referred to as the 1st, 2nd, and 3rd electrostatic breakdown (1st BD, 2nd BD, and 3rd BD), respectively.

These three discharge domains were verified in finite element method simulations. For the short-circuit condition, as the surface charge density on TL $\sigma_{TL}$ increases, electric field intensities at the bottom of CCE and beside the corner of FE increase until the threshold of the electrostatic breakdown is reached (Fig. 4e). In an open-circuit condition, the charge generated by the friction cannot be efficiently

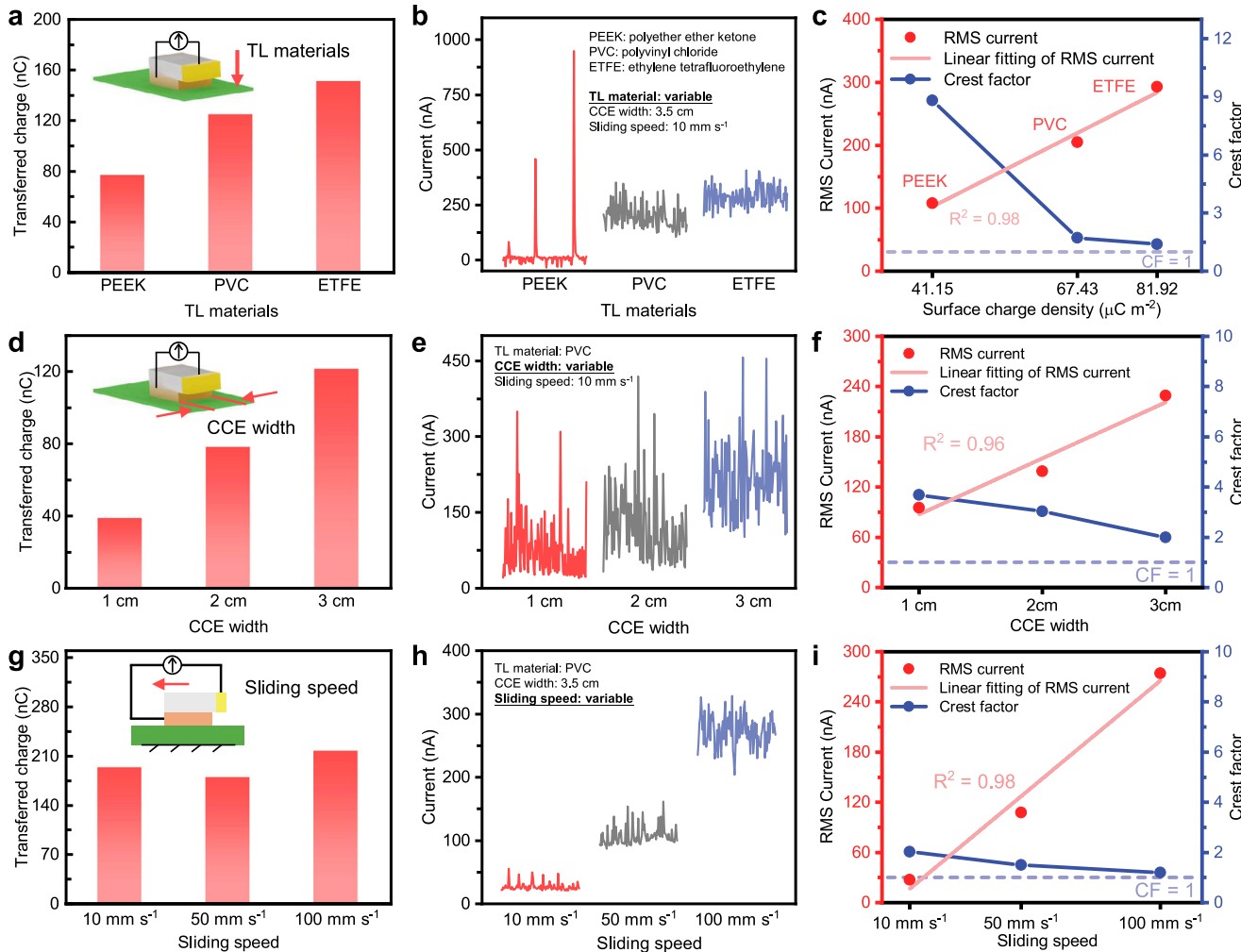

**Fig. 3 | Output performance of DC-TENG under different conditions.** With 3.5 cm CCE width and 10 mm s⁻¹ sliding speed, **a** Transferred charge, **b** Output current, and **c** RMS currents and CFs with different materials serving as TL. With PVC serving as TL and 10 mm s⁻¹ sliding speed, **d** Transferred charge, **e** Output current, and **f** RMS currents and CFs with different CCE widths. With PVC serving as TL and 3.5 cm CCE width, **g** Transferred charge, **h** Output current, and **i** RMS currents and CFs with different sliding speeds. Source data are provided as a Source data file.

transferred and thus will accumulate on FE, introducing a dramatically high potential difference between FE and CCE, which triggers the electrostatic breakdown between the two electrodes (Fig. 4f). Here, open-circuit conditions include large resistive loads, capacitive loads, and completely open circuit conditions, since they have similar electrical performance. Figure 4g shows the finite element method simulation results on the sample point at 5 μm above the corner of FE, the sample point at 5 μm on the lower right corner of CCE, the discharging path between FE and CCE, and the discharging path between CCE and TL. Supplementary Fig. 9 and Supplementary Note 13 explain the sample point selection principle in finite element method simulation in more detail. The simulation results suggest that the area between the CCE and TL is not the only discharge domain.

To intuitively validate the existence of electrostatic breakdown at specific positions, observation experiments were designed and conducted. A DC-TENG was fixed with FE facing upwards and a transparent PVC film adhered on the acrylic substrate serving as TL. A camera was held above the whole device to take pictures from a top view (Fig. 4h). Since the brightness of the corona discharge is too low to be photographed, a capture with 30 s exposure was applied. As TL rubbed FE back and forth, in addition to the conventional breakdown in the 1st and 2nd domains, there is also another 2ⁿᵈ domain breakdown in the opposite direction, which is the luminous line on the left (Fig. 4i). A

photo of the experimental device is provided in the lower right inert in Fig. 4i. The same experimental setup, but with an open-circuit electric configuration, was used to observe discharges in the 3ʳᵈ domains. The image in Fig. 4j was taken at the moment of the electrostatic breakdown happening. Since there is no such electric arc when the output of the external circuit is normal, it is not the luminescence caused by the 1st breakdown; since this arc still exists after the 2ⁿᵈ domain is blocked (Supplementary Fig. 10), it can only be caused by the 3ʳᵈ breakdown. To increase the brightness of the light emitted by the electric discharge, a capacitor was connected in parallel in the circuit, as shown in the top inset. Moreover, electrical experimental data also provides evidence of electrostatic breakdown occurring at different locations as described above. The breakdown in the 2ⁿᵈ domain under short circuit condition can be inferred by comparing the total triboelectric charge with the charge participating in the 1st BD (Fig. 4k). The breakdown in the 3ʳᵈ domain under open-circuit condition can be validated by charging a capacitor and observing the sudden dropdown (Fig. 4l). More analysis can be found in Supplementary Figs. 11, 12 and Supplementary Note 14–16.

A DC-TENG with split FEs was provided in this paper to further understand, quantify, and regulate the 2ⁿᵈ BD. Instead of using a single piece of copper foil as FE, the copper foil is cut into two pieces and connected with an electrometer. A piece of Kapton tape, due to its

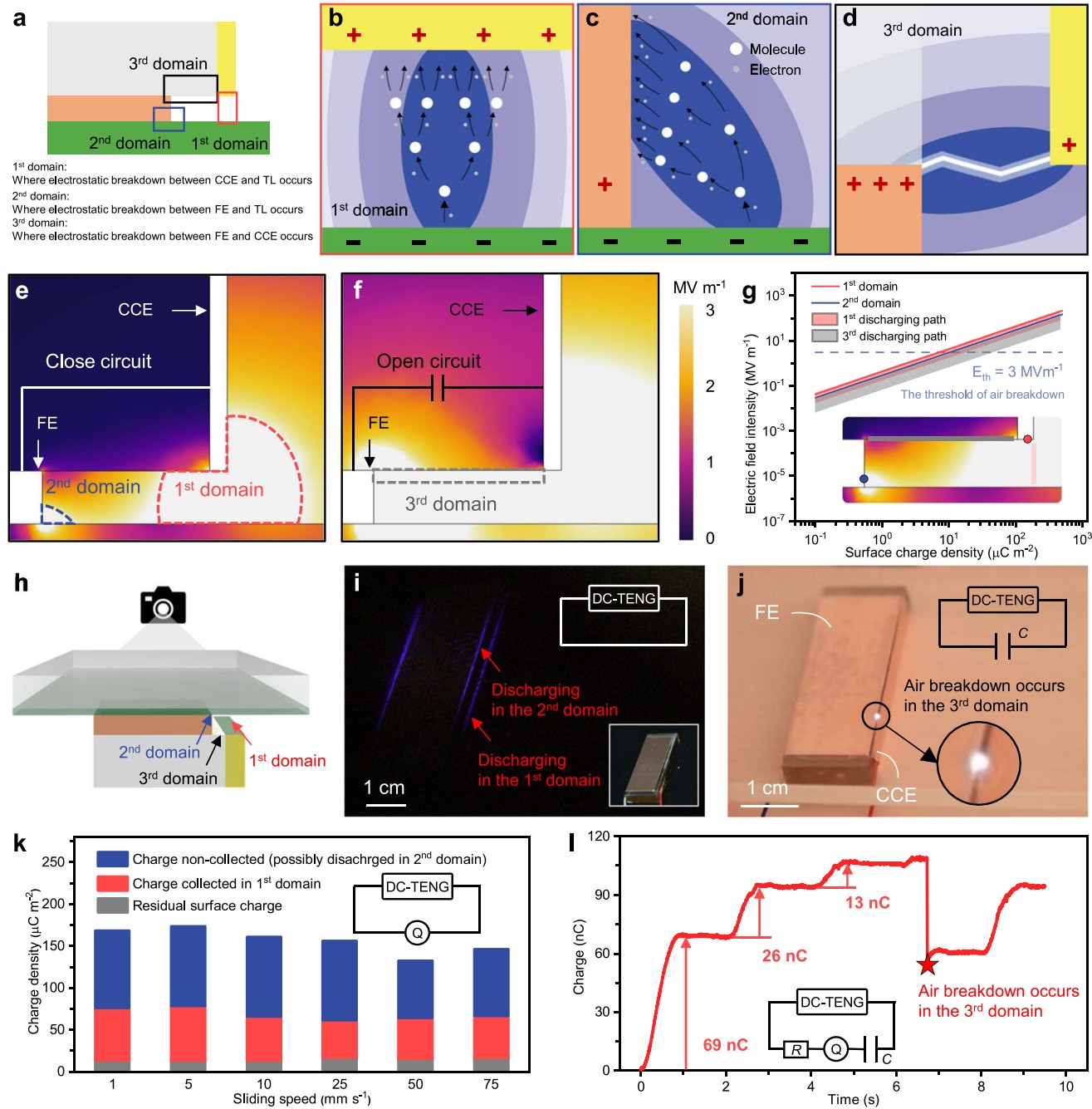

**Fig. 4 | Definition and evidence for the three main discharge domains.**
**a** Definition for the three discharge domains. **b** Schematic diagram for the discharge in the 1st domain. **c** Schematic diagram for the discharge in 2nd domain. **d** Schematic diagram for the discharge in the 3rd domain. **e** Simulated results of the discharge occurs in the 1st and 2nd domains. **f** Simulated results of the discharge occurs in the 3rd domain. **g** Simulated results of the electric field intensity in discharge domains with different surface charge densities. **h** Hardware settings of the discharge observation experiments. **i** Photo for the discharge in the 1st and 2nd domains (exposure with 30 s). **j** Photo for the discharge in the 3rd domain. **k** Total triboelectric surface charge density, transferred surface charge density, and residual surface charge density measured and calculated under different sliding speeds. **l** Transferred charge measured when charging a capacitor. Source data are provided as a Source data file.

excellent insulation properties, is used to isolate the two sections, defined as an isolation layer, which makes the charge transfer inside FE able to be measured. The isolation layer has little influence on contact electrification and output signals, as discussed in Supplementary Fig. 13 and Supplementary Note 17. The working mechanism is described by four steps, as shown in Fig. 5a. The contact between FE and TL generates charges on the surfaces of FE, TL, and the isolation layer (Fig. 5a(i)). As FE and TL relatively move towards each other, the charged triboelectric surface moves behind FE, creating a sufficiently high electric field

intensity to induce the 2nd BD (Fig. 5a(ii)). This segment of TL with a reduced surface charge density due to the 2nd BD then moves under CCE and results in the 1st BD (Fig. 5a(iii)). When the relative motion ends, the electric output in the external circuit stops (Fig. 5a(iv)). It is worth noting that when a conventional DC-TENG works, the 2nd BD is an inevitable accompaniment to contact electrification on FE, leading to waste and loss of triboelectric charge. The DC-TENG with split FEs can not only quantify the charges participating in the 1st and 2nd domains respectively by splitting the two processes (contact electrification and

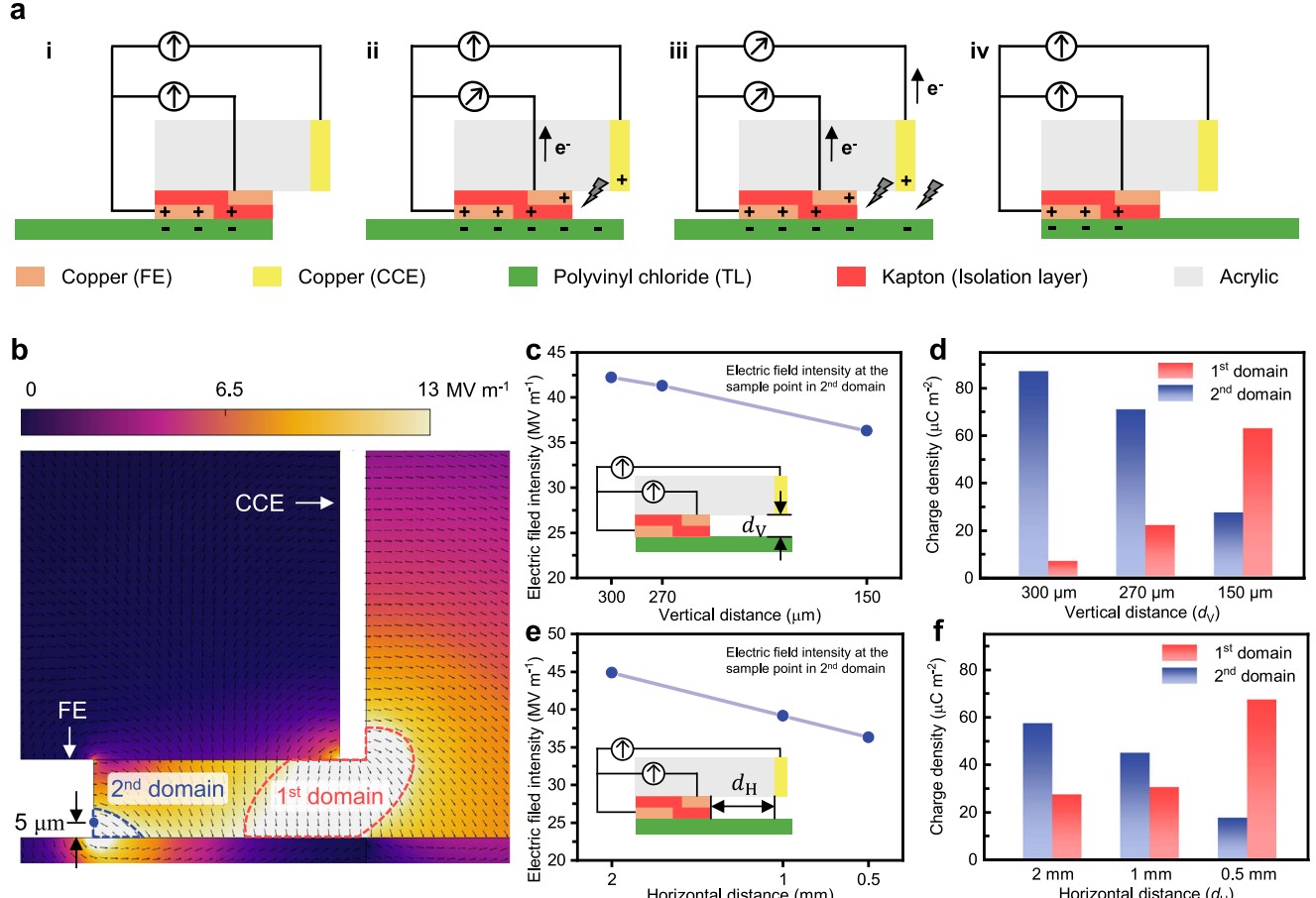

**Fig. 5 | Regulation on the 1st and 2nd BDs. a** Working mechanism of the DC-TENG with split FEs for quantifying the charges in the 1st and 2nd domains. **b** Electric field intensity distribution presented by finite element method simulation. **c** Simulated results of the electric field intensity at the sample point in the 2nd domain under different vertical distances (the inset indicates the position of the variable vertical distance $d_V$). **d** Charge participating in the 1st and 2nd electrostatic breakdowns under

different vertical distance $d_V$ conditions. **e** Simulated results of the electric field intensity at the sample point in the 2nd domain under different horizontal distances (the inset indicates the position of the variable vertical distance $d_H$). **f** Charge participating in the 1st and 2nd electrostatic breakdowns under different horizontal distance $d_H$ conditions. Source data are provided as a Source data file.

the 2nd BD) on FE, but also be a solution for boosting the output performance, since it can make use of both the charges transferred inside conventional FE and the charges transferred between FE and CCE.

The electric performances on both the external circuits between the two portions of FE and between FE and CCE were simulated and tested as the vertical and horizontal distances between FE and CCE varied. The finite element method simulation configuration is shown in Fig. 5b, where the blue circle is the sampling point in the 2nd domain. As shown in Fig. 5c and Supplementary Fig. 14, electric field intensity at the sampling point in the 2nd domain decreases with a decrease in vertical distance between FE and CCE, indicating that a smaller distance suggests a weaker electric field intensity and less amount of charge has to be released to make the electric field intensity in the 2nd domain lower than the breakdown threshold, so more charge left on the surface of TL and can be released in the 1st domain. To confirm it, the vertical distance was varied by attaching an isolation layer or FE layers of different thicknesses and experimental data (Fig. 5d) shows the exact trend that with a smaller vertical distance, more charge participates in the 1st BD. The amount of charge participating in the 1st and 2nd BDs can also be regulated by changing the horizontal distance (Fig. 5e and Supplementary Fig. 15). With simulated results and experimental data provided in Fig. 5c–f, it is obvious that a smaller vertical or horizontal distance between FE and CCE result in more charges participating in the 1st BD, which implies that more charge can be employed to do work in the external circuit, and thus a higher electric output can be achieved.

## Comprehensive "cask model" and optimization method of DC-TENG

Concluded from the statement above, a comprehensive model, "cask model", of DC-TENG is proposed. As illustrated in Fig. 6a, a triboelectric power generating system can be considered as a container, and charges inside the system can be regarded as the working fluid in the container. As the contact electrification process propels charges into the system, the stored charge density increases (Fig. 6a(i)). When the breakdown threshold between the conductive electrodes and the dielectric layer is reached, the 1st and 2nd BDs are triggered and current output in the external circuit occurs (Fig. 6a(ii)). Based on this dynamic description, the output current can be significantly increased if the 2nd breakdown is efficiently reduced or even eliminated (Fig. 6a(iii)).

Different from the cascaded-capacitor-breakdown model presented under ideal conditions, the inlet charge for this "cask model" is the total triboelectric charge generated by the friction between FE and TL (sum of three kinds of charges in Fig. 4k). The output of "cask model" takes the three discharge domains as well as the residual charge into account. For the 3rd outlet, since the resistance in the external circuit between FE and CCE is generally not large enough to block the charge transferred between the two electrodes, the potential difference between FE and CCE is far away from the breakdown voltage, which means triboelectric charge can be rarely released from this outlet. For special applications, facing the risk of electrostatic breakdown in the 3rd domain, reducing EFI between FE and CCE or increasing the dielectric

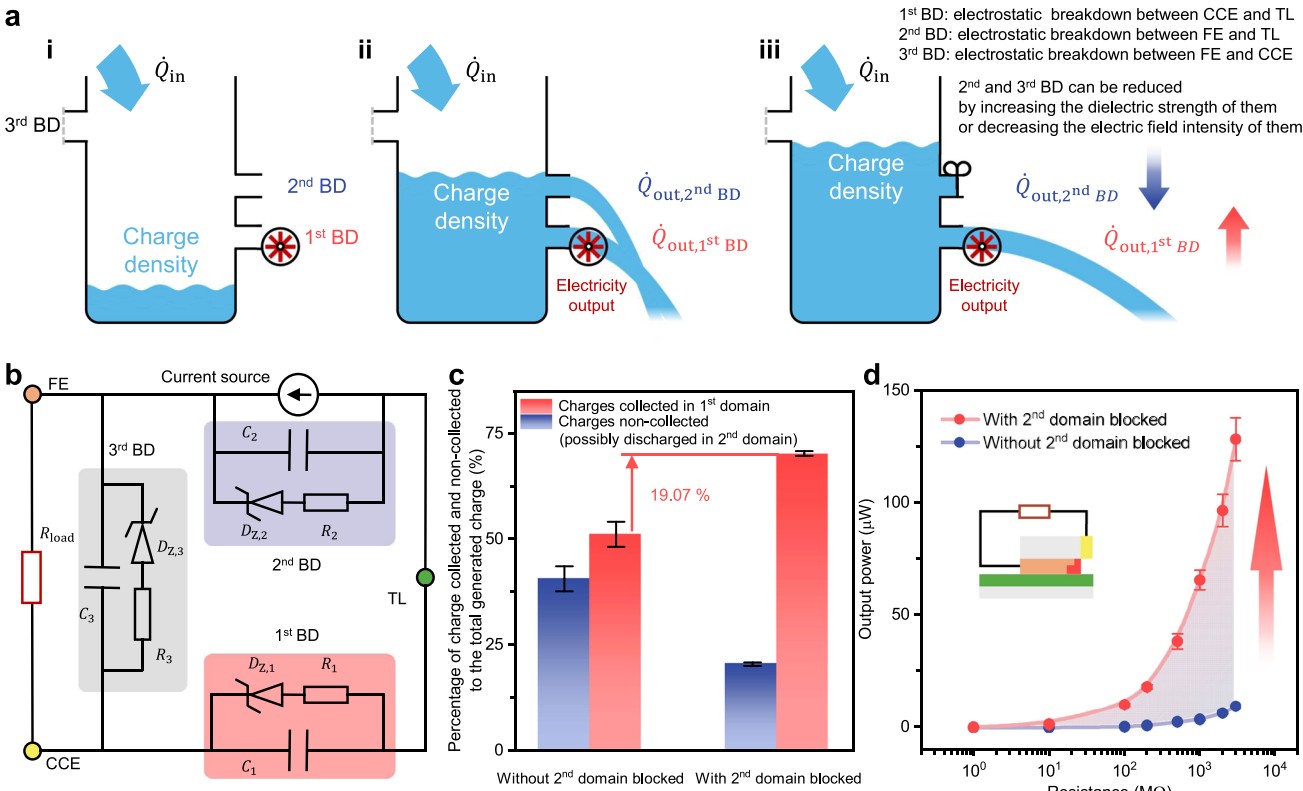

**Fig. 6 | "Cask model" and optimization. a** "Cask model" for DC-TENG. **b** Equivalent circuit for DC-TENGs. **c** Percentage of charge participating in the 1st electrostatic breakdowns to the total triboelectric charge with and without the 2nd domain blocked. **d** Output power under different external load conditions with and without the 2nd domain blocked (the inset is the schematic diagram for the DC-TENG with end-isolated FE). Error bars represent standard deviation with five trials. Source data are provided as a Source data file.

strength between FE and CCE, for example, by enlarging the distance between the two electrodes or filling the space between FE and CCE with dielectric materials work. To better understand it in a circuit, an equivalent circuit is shown in Fig. 6b. The electric properties in the 1st, 2nd, and 3rd domains are similar, so FE and CCE, FE and TL can also be analyzed as capacitor models similar to the one in Fig. 1c. All possible discharge behaviors around DC-TENG can be regarded as the 1st discharge has been copied to the other two domains in some way. More notes on this equivalent circuit can be found in Supplementary Note 18.

As both Fig. 6a, b demonstrate, increasing the dielectric strength of the 2nd domain, which makes it difficult to trigger the 2nd BD, is a feasible method to increase the output power. Based on Fig. 4i, the 2nd BD mainly occurs at the edge of FE close to CCE side. To prevent the 2nd BD, a DC-TENG with end-isolated FE was designed, for which the space between FE and CCE is filled with Kapton tape instead of being left with air. The Kapton here acts as an isolation layer. Since the dielectric strength of Kapton is higher than that of the air, it requires much higher electric field intensity to trigger the 2nd BD, which means it is difficult for charges to leave from the 2nd domain, resulting in more charges releasing at the 1st BD. Experimental results are shown in Fig. 6c, d, and the inset demonstrates the configuration for avoiding the 2nd breakdown. With five sets of repeated experiments, the percentage of the amount of charge transferred between FE and CCE to the total amount of triboelectric charge was measured and calculated in a close circuit. It is obvious that more charges entered the external circuit by blocking 2nd breakdown with Kapton tape and thus the output power under different external load boosted. Under 1.414 Hz motion frequency, in the wide resistance range of 1 MΩ to 3 GΩ, the output power increased by over 12.5 times to 40 times by blocking the 2nd domain (Supplementary Table 1). More details on experiments are shown in Supplementary Fig. 10 and Supplementary Note 19.

The experimental results not only validate "cask model" mentioned above, but also provide a feasible solution (blocking the 2nd breakdown) to increase the amount of charge released in the 1st domain and thus greatly improves the output performance under a wide range of resistive loads.

## Discussion

In summary, going beyond the limited discharge region previously assumed, we identified and regulated the total three discharge domains of DC-TENG, which filled the gap between the ideal theoretical model and the actual circumstances. By analyzing the equivalent capacitor $C_{CCE}$, a cascaded-capacitor-breakdown model was built for the ideal condition and confirmed with experiments, revealing that under a relatively constant environmental condition (Supplementary Note 20), the main factors that influence the output performance of DC-TENGs are $\sigma_{CCE}$, $w_{CCE}$, and $v_{sliding}$, since all of them are linearly related to the dynamic output. This model only considers charges collected at CCE (i.e., the assumption that there is only one discharge domain in the operation of the DC-TENG), which is actually significantly smaller than the total triboelectric charges. Further evidence from finite element method simulation, observation experiments, and electric signals suggested there are three discharge domains. "Cask model" concluding the three discharge domains and the regulation method for discharge domains were thus presented. Based on that, a specific method for decreasing the charge loss in the 2nd domain and increasing the dynamic output was proposed, achieving an order of magnitude more power output than that of the conventional DC-TENG in a wide range of resistive loads.

For better understanding and future applications, the concept of the discharge domain for a conventional DC-TENG can be further generalized (Supplementary Fig. 16 and Supplementary Note 21). The

generalized definitions for discharge domains are: the 1st domain is the region where the discharge on the conductor between the electronics in the external and the bottom of CCE locates; the 2nd domain is the region where the discharge on the conductor between the FE-TL contact interface and the electronics locates; the 3rd domain is the region where discharge between the two conductive electrodes locates. Thus, all the potential discharge locations are defined into three categories due to their characters and influence on the output performance. With the systematic analysis on discharge domains of DC-TENG, more fundamental theories about contact electrification and electrostatic discharge[36,39] can be utilized to understand and further improve its performance, which previously relied on empirical observations and phenomena. Furthermore, the models and results in this paper may also be applied to tribology, triboelectrification, and other electrostatic phenomena. In a word, the "cask model" introduced in this work not only points out the direction for substantially increasing and regulating outputs, opening a new chapter of DC-TENG dynamic outputs, but also provides insights into the underlying physical mechanisms, ultimately leading to a more comprehensive understanding of this field.

## Methods

### Fabrication of the conventional DC-TENG for confirming the cascaded model of DC-TENG

Three sets of DC-TENG devices were fabricated for confirming the influence of $\sigma_{CCE}$, $w_{CCE}$, and $v_{sliding}$, respectively. For the variable $\sigma_{CCE}$, a DC-TENG device composed of FE (3.5 cm × 1.5 cm × 50 µm) and CCE (3.5 cm × 0.5 cm × 50 µm) made of copper foil was fabricated. A substrate acrylic block was cut using a laser cutter (PLS6.75, Universal Laser System) and pre-adhered with a piece of Kapton tape (50 µm of thickness) to make the surface smoother. FE and CCE were then attached on the bottom and lateral surface, respectively, with a horizontal distance of 0.5 mm and a vertical distance of 100 µm. Three TLs with different materials were made by adhering a foam layer on acrylic substrates, followed by a layer of PEEK, PVC, or ETFE film (50 µm), respectively. For $w_{CCE}$, first of all, a DC-TENG device composed of FE (3 cm × 1.5 cm × 50 µm) and CCE (3 cm × 0.5 cm × 50 µm) was fabricated with other settings same to that mentioned above. After the experiments for the width of 3 cm, FE and CCE were cut into size 2 cm × 1.5 cm × 50 µm and 2 cm × 0.5 cm × 50 µm to do the experiments for the width of 2 cm. Similar operations were also done to achieve 1 cm width experiments. For $v_{sliding}$, a basic DC-TENG device composed of FE (3.5 cm × 1.5 cm × 50 µm) and CCE (3.5 cm × 0.5 cm × 50 µm) made of copper foil was fabricated with the same process mentioned above.

### Fabrication of the DC-TENG with split FEs

A square acrylic plate (5 mm of thickness) was cut with a laser cutter (PLS6.75, Universal Laser System) and pre-adhered with a piece of Kapton (50 µm of thickness) tape to make the surface smoother. A piece of copper foil (3.5 cm × 0.5 cm × 50 µm) was adhered on the lateral surface of the substrate as CCE. A stripe of copper foil (3.5 cm × 2.0 mm × 50 µm) was adhered on the bottom surface of the substrate as one part of FE, 0.5 mm horizontally away from CCE. A square piece of Kapton tape (3.5 cm × 1.5 cm × 50 µm) was adhered on both the bare surface of the substrate and the FE part as the isolation layer. A square of copper foil (3.5 cm × 1.3 cm × 50 µm) was adhered on the Kapton tape as another part of FE, 2.5 mm horizontally away from CCE. TL was made by adhering a foam layer on acrylic substrates (5 mm), followed by a layer of PVC (50 µm). Two DC-TENGs were made of this basic configuration for the experiments of variable vertical distance and horizontal distance, respectively. For the vertical distance tests, after the experimental data was collected by the basic device, a piece of Kapton tape (20 µm) was adhered to the bottom of the whole device and the same settings of FE and the isolation layer were done to achieve the total vertical distance of 270 µm. The experiments of

300 µm were achieved by adhering Kapton tape of 50 µm thickness and the same settings of FE and the isolation layer onto the basic device after tearing off the other materials. For the horizontal distance tests, after the experimental data collected by the basic device, the first part of FE together with the isolation layer on it was cut to achieve 1 mm and 2 mm horizontal distance between FE and CCE, so that the consistency of this series experiments can be guaranteed.

### Fabrication of the DC-TENG with end-isolated FE

A square acrylic plate (5 mm of thickness) was cut with a laser cutter (PLS6.75, Universal Laser System) and pre-adhered with a piece of Kapton (50 µm of thickness) tape 2 mm horizontally away from CCE. A piece of copper foil (3.5 cm × 1.35 cm × 50 µm) was 1 mm horizontally away from CCE, adhered on the bare bottom surface of the substrate and the Kapton tape as FE. A stripe of Kapton tape (4 cm × 1.5 mm × 20 µm) was 0.5 mm horizontally away from CCE, adhered on FE as the isolation layer. TL was made by adhering a foam layer on acrylic substrates (5 mm), followed by a layer of PVC (50 µm).

### Environmental setting

To minimize the influence of environmental factors on the experimental results, the temperature and relative humidity were carefully controlled in a laboratory environment. Unless otherwise specified, all experiments were conducted at a temperature range of 293.15–298.15 K and a relative humidity level of 10%–20%.

### Characterization

The output current and transferred charge were measured by an electrostatic electrometer (Keithley 6514) and the signals were real-timely acquired with NI-6218 and its corresponding LabVIEW controlling software. Relative sliding motions in this paper were achieved by a linear motor (TSMV120-1S, LinMot). Photos in this paper were taken with a digital camera (Nikon D750). Finite element method simulation was down in COMSOL Multiphysics software.

## Data availability

The authors declare that all the data that support the findings of this study are available within the article and its supplementary information files. Source data are provided with this paper.

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

## Acknowledgements

Research was supported by the National Key R & D Project from Minister of Science and Technology (2021YFA1201602), National Natural Science Foundation of China (Grant No. U21A20147 and 62204017), China Postdoctoral Science Foundation (2021M703172), Innovation Project of Ocean Science and Technology (22-3-3-hygg-18-hy), and the Fundamental Research Funds for the Central Universities (E1E46802).

## Author contributions

J. Zhang, Y.G., D.L., J. Zhao, and J.W. conceived the idea. J. Zhang and Y.G. performed the experiments and analyzed the data. J. Zhang and D.L. drafted the paper. J. Zhao and J.W. revised the paper. D.L., J. Zhao, and J.W. supervised this work. All the authors discussed the results and commented on the manuscript.

## Competing interests

The authors declare no competing interests.
