## [Peer Review File · Nature Communications]

Discharge domains regulation and dynamic processes of direct-current triboelectric nanogeneratorREVIEWER COMMENTS

Reviewer #1 (Remarks to the Author):

In this paper, the authors tried to simulate and verify models to explain the electricstatic discharge in DC-TENG. An enriched discussion has been made. The information that afford here could help researcher to better understanding the discharge process in a DT-TENG. However, the hypothesis in some the model did not match the real case of a DC-TENG. The guidance to develop higher performance DC-TENG is limited. Generally, the paper afford in-depth information about the discharge in DC-TENG, but the models need to be improved and the significance is not so obvious.

- 1) Electrostatic discharge at domain 3 actually does not exist in real case, as the FE and CCE are connected. It only happens when FE and CCE are not connected, which not a case of a DT-TENG.
- 2) The section "Cascaded-capacitor-breakdown model and experimental validation under ideal conditions" does nor afford better understanding than the published paper: Liu et al., Sci. Adv. 2019; 5 : eaav6437.
- 3) Two many abbreviations, making it hard to follow the context.
- 4) As shown in Figure 6d, it seems the kapton IL attend the triboelectrification process. If it was case, then the output power enhancement will not due the isolation of the kapton, but the triboelectric property. Copper and kapton have different charge affinity, and the combination as shown in the figure, will definitely increase the output. That said, the conclusion is questionable.
- 5) The "Cask model" that presented in Figure 6 need to be optimised. The 3rd BD, does not exist if FE and CCE are connected. Even the IL works, it will increase the speed of charge accumulation but not the level, because the discharge happen automatically when potential achieved to a certain level.
- 6) Some abbreviation is not defined, e.g. FE.

Reviewer #2 (Remarks to the Author):

The manuscript NCOMMS-23-02049 "Discharge domains regulation and dynamic processes of direct-current triboelectric nanogenerator" proposes a new model to fully understand direct-current triboelectric nanogenerator (DC-TENG). This Reviewer does not have much concerns about the technical aspects and the electrical measurements, but some fundamental aspects regarding the mechanisms by which surfaces build-up charges (and the discussion section) need clarification. Also, the paper seems biased on TENGs and the real challenge here is to apply this model/results not only for DC-TENG, but also for tribology, triboelectrification and other electrostatic phenomena. I suggest that the authors address the following questions/concerns:

- 1) The generation of (only) opposite charges on FE and TL upon friction (or contact) is not supported by many important papers. In fact, bipolar, multipolar and mosaics of charges on tribo surfaces have been imaged and identified at both micro and macro scales. Thus, the first paragraph in the results section and Figure 1/Figure 4/Figure 5 are not accurate since the flow of charges and the net charge are different from charge domains. If this is the case, authors must provide additional results. For example, this claim could be supported by electrostatic mappings of FE and TL done with macroscopic Kelvin probes. Charge domains could also be revealed with a charged tonner (e.g. Xerox® Cyan Developer – powder iGen3, 5R706) or other means. Kelvin probe force microscopy can also help the discussion, but only as a complementary tool since charge domains are macroscopically distributed.
- 2) Some of the work reported here is similar to the work reported on DOI: 10.1002/anie.201806658 to indicate the effects caused by distant substrates on the outcome of contact electrification. Also, a recent paper (DOI: 10.1038/s41567-022-01714-9) deals with

electrostatic charging during peeling, showing that charge domains are formed not due properties of the contacting materials but due to electrostatic discharges between the separating surfaces. In fact, sparks not only dissipate the surface charges but invert their polarities at the ends of the spark. Thus, electric current oscillations reported in the manuscript could have the same origin as those reported in the mentioned papers.

3) How the surface charge density is effectively calculated? If the authors are measuring the transferred charge using the Coulomb function (and/or the current function through Keithley 6514 triax cable high input) this is not surface charge density. For example, charges must flow through the junctions of contact at the interface developed due asperities of both contacting surfaces. In other words, electric charge is being transferred at the real area of contact, usually less than 1% of the apparent contact area. Even thermal fluctuations at the metal-dielectric interface during sliding contact can affect the measuring charges. Again, electrostatic mappings would bring rich information to this manuscript.

Some minor points:

1) The effect of rougher TL surfaces should be discussed, as contact junctions are important to contact electrification and may also play an important role in the cascaded-capacitor-breakdown model.

2) A triboplasma is frequently generated under sliding contact conditions. Thus, triboemission and triboluminescence processes could be occurring while electrical charges ionize the surrounding air, which would have the same outcome registered in Figure 4. Thus, the claim that corona discharge is occurring at the interface needs further verification. This could be confirmed by conducting the experiments in both neon gas atmosphere and under high vacuum ($< 10^{-7}$ bar).

3) With the experimental conditions mentioned before, authors can provide Paschen plots. In fact, recent works have shown that for vertical distances in the micrometer range, the breakdown voltage shows a significant reduction. Moreover, it is very unusual that the Paschen law is not even mentioned in a paper claiming that the results are explained through an electrostatic breakdown model.

4) What was the relative humidity and temperature conditions? Adsorbed water plays a key role on both charging and discharging phenomena. Actually, water that is frequently described as playing a passive role on triboelectrification (increasing surface conductivity), but many papers have been describing its active role on charging and induction. Authors must discuss this within the text.

5) The paper is full of acronyms and any reader gets really tired after the first three pages. Authors must bring another solution for this issue. Also, the FE acronym is not explained through the text. From Figure 1, I am assuming that "FE" is the "Friction Electrode".

6) The text has few typos.

Reviewer #3 (Remarks to the Author):

The manuscript has introduced cascaded-capacitor breakdown model for DC-TENGs. Considering current understanding of DC-TENG is mostly focused on breakdown mechanism on each electrode, the proposed mechanism seems promising and the experimental results match with simulation data and theoretical analysis. However, there are some comments to be addressed before I can recommend this paper to be published. Detailed comments are as below:

1. Considering the manuscript is mainly focused on analyzing the working mechanism of DC-TENG, it should include description and appropriate reference of paschen's law and field emission during breakdown process. I believe that explanation on paschen's law and field emission would give in-depth explanation why the electric breakdown occurs as vertical distance and horizontal distance of DC-TENG changes.

2. In Figure 5a, the authors have ignored the influence of contact electrification between kapton tape and PVC film. However, since the electrons from the PVC surface constantly flow to electrodes of 1st and 2nd domain, there seems to be constant loss of electrons on PVC surface and would lead to contact electrification of kapton tape and PVC during operation. In addition, as kapton tape is dielectric material, there would be dielectric loss when surface charge of kapton film influences charge of FE through electrostatic induction. This could lead to electrical potential difference between two electrodes of FE part separated with kapton film. The authors should show transferred charge data between two electrodes of FE without the CCE connected to ensure that there is no electron flow (electrical potential difference) between them.

3. The authors should show matching resistance of the DC-TENG since it is important parameter for application of TENGs. Figure 6d only shows the output power of EIFE-DC-TENG increasing depending on external resistance.

4. In addition, considering conventional TENG has matching resistance around 10 to few hundred megaohm, the EIFE-DC-TENG seems to have quite high matching resistance over 3 gigaohm. Can authors explain the reason why it shows such high matching resistance?

Point-by-point responses to the reviewers' comments

We sincerely thank the reviewers for carefully reviewing our work, which are indeed very helpful to make the paper more solid and smooth. We have revised our manuscript in accordance with your pertinent comments and suggestions. The following responses are prepared to address all of the reviewers' comments in a point-by-point fashion. (Comments in black, responses in Blue, revisions in Red.)

REVIEWER COMMENTS

Reviewer #1 (Remarks to the Author):

In this paper, the authors tried to simulate and verify models to explain the electrostatic discharge in DC-TENG. An enriched discussion has been made. The information that affords here could help researcher to better understanding the discharge process in a DT-TENG. However, the hypothesis in some the model did not match the real case of a DC-TENG. The guidance to develop higher performance DC-TENG is limited. Generally, the paper affords in-depth information about the discharge in DC-TENG, but the models need to be improved and the significance is not so obvious.

Response:

We highly appreciate the reviewer for carefully reviewing our work, and thank your valuable comments on our research work.

1. Electrostatic discharge at domain 3 actually does not exist in real case, as the FE and CCE are connected. It only happens when FE and CCE are not connected, which not a case of a DT-TENG.

Response:

Thank you very much for your careful review.

As a power source, though the direct-current triboelectric nanogenerator (DC-TENG) can directly power electronics and store energy, to efficiently transfer the output energy of DC-TENG into the required energy of electronics, it is better to equip the DC-TENG with a power management system or energy storage system. However, these systems involve complex external circuits, which are equivalent to a large load when

they work and make the electrostatic breakdown in the 3rd domain prone to be triggered. Additionally, to achieve higher output power density, microstructure design is often utilized¹, which requires a small distance between the friction electrode (FE) and the charge collecting electrode (CCE). As the distance decreases, the likelihood of the 3rd breakdown occurring increases, limiting the further increase of output power. Therefore, the 3rd breakdown must be considered for future high-output DC-TENG design. Moreover, compared to other mechanical energy harvesting technologies (such as piezoelectric nanogenerators), DC-TENG inherently has high-voltage output characteristics, making it promising for high-voltage applications. Thus, it is more prone to experiencing the 3rd breakdown in high-voltage scenarios. Overall, electrostatic breakdown in the 3rd domain is common in DC-TENG applications, and is critical to the continued development of DC-TENG.

Here, we proposed two examples to illustrate the occurrence of the 3rd breakdown from the perspective of resistive and capacitive loads. These examples are based on the device of a relatively large distance between FE and CCE. When the distance between FE and CCE is reduced, the breakdown voltage decreases according to Paschen's law, namely the likelihood of the 3rd breakdown occurrence increases.

Calculated from the experimental data in **Fig. 6**, for a conventional DC-TENG, the voltage between CCE and FE is around 60 V under a load of 1 G Ω , 110 V under a load of 2 G Ω , and 160 V under a load of 3 G Ω . For the DC-TENG with end-insulated FE, the voltage reaches 250 V at 1 G Ω , 430 V at 2 G Ω , and 610 V at 3 G Ω (**Fig. R1**). Notably, the 3rd electrostatic breakdown occurs when the external load is 6 G Ω . **Fig. R2** illustrates the output current under this condition, with the 3rd breakdown manifesting as a sudden drop. Besides the applications involving resistive load applications, the significance of the 3rd breakdown is also emphasized in scenarios featuring capacitive loads. **Fig. R3** exhibits a sudden drop in a capacitor charging process and **Fig. R4** shows the electric arc emitted between the two electrodes. Both are attributed to the occurrence of the 3rd electrostatic breakdown. When the 1st breakdown occurs, the released charges enter the electrometer through CCE, and the amount of charge passing through the electrometer continues to grow. When the

potential difference between the two electrodes is large enough, the charge suddenly drops, and electric arc forms. Even with the second domain blocked, the bright arc still occurs (Fig. R5). Comprehensive considered, these phenomena are due to the 3rd breakdown.

Figure R1 Output voltage of conventional DC-TENG and DC-TENG with end-isolated FE under high resistive loads.

Figure R2 (Supplementary Fig. 10c) Discharge signal on the current output when the 3rd breakdown occurs.

Figure R3 (Fig. 4) Transferred charge measured when charging a capacitor with a conventional DC-TENG.

Figure R4 (Fig. 4j) Light emitted by the 3rd breakdown.

Figure R5 (Supplementary Fig. 10b, the left insert is Supplementary Fig. 10a) Evidence for the 3rd breakdown occurs on the DC-TENG with the 2nd domain blocked.

In summary, the occurrence of the 3rd breakdown is a tangible possibility in practical scenarios, particularly in a power management system or energy storage system which necessitates large resistive loads and capacitive loads. Accordingly, to holistically examine the discharge phenomena on DC-TENGs and to offer guidance for real-world applications, the electrostatic discharge within the 3rd domain has to be taken into consideration.

In the original manuscript, the term “open-circuit” denotes a simplified

representation of scenarios involving large resistive loads, capacitive loads, and completely open circuits. The absence of an elaborate explanation of this term may confuse readers, and thus, we have included a more detailed description in the revised manuscript. Additionally, we have further validated the existence of the 3rd breakdown and provided a better understanding of the discharge phenomena on DC-TENG.

The revised part in the present manuscript is as follows:

We have made a revision of “In the working process of DC-TENG, this kind of discharge typically arises in instances of high resistive or capacitive loads within the external circuit, as well as when the DC-TENG is employed as a high-voltage source.” in the first paragraph of “**Identification and regulation of three discharge domains**”.

We have added a description of “...which triggers the electrostatic breakdown between the two electrodes (**Fig. 4f**). Here, open-circuit conditions include large resistive loads, capacitive loads, and completely open circuit conditions, since they have similar electrical performance.” in the second paragraph of “**Identification and regulation of three discharge domains**”.

We have made a revision of “The image in **Fig. 4j** was taken at the moment of the electrostatic breakdown happening. Since there is no such electric arc when the output of the external circuit is normal, it is not the luminescence caused by the 1st breakdown; since this arc still exists after the 2nd domain is blocked (**Supplementary Fig. 10**), it can only be caused by the 3rd breakdown.” in the third paragraph of “**Identification and regulation of three discharge domains**”.

2. The section "Cascaded-capacitor-breakdown model and experimental validation under ideal conditions" does not afford better understanding than the published paper: Liu et al., Sci. Adv. 2019; 5 : eaav6437.

Response:

Thank you very much for your careful review.

There is no doubt that our previous findings and perspectives are of significant importance and hold a great impact on related fields. It proposed a basic model for the direct-current triboelectric nanogenerator (DC-TENG), emphasizing its capacity to

generate constant-current output signals. However, our present work, as detailed in the section entitled "Cascaded-capacitor-breakdown model and experimental validation under ideal conditions", focuses on quantitatively comprehending the DC-TENG, determining the variables that have a significant influence on attaining constant-current output, providing guidance for the future design aimed at the constant-current output. We built a more detailed model with two practical parameters: the random factor for the contact electrification process and a cascaded simulation structure in the electrostatic discharge process. By considering the three main variables mentioned in the manuscript, we developed a detailed numerical simulation model to calculate the dynamic output of a DC-TENG, which is consistent with experimental results. In practice, according to the output performance requirements of different application scenarios, our model can guide materials selection, structural design, and working parameter settings for DC-TENG operations, especially for those requiring constant-current output.

More importantly, the purpose of this section is to provide a basic example for all three discharge domains around the DC-TENG, as described in the following two sections. In this section, the analysis and numerical description of the electrostatic discharge in the 1st domain deepen our understanding of the electrostatic discharges in DC-TENG working conditions. When the discharge locations are expanded to three, as the proposed "cask model" illustrates (**Fig. 6a and b**), we can regard it as the discharge in the 1st domain has been copied to the other two locations in some way. Notably, the section "Cascaded-capacitor-breakdown model and experimental validation under ideal conditions" described and validated a model under ideal conditions, which only holds for the common acceptance of previous studies (except for a small number of charges remaining on triboelectric layer (TL) surface, triboelectric charges produced at friction electrode (FE) are electrostatically broken-down at charge collecting electrode (CCE)). The real-world condition, namely the total of three discharge domains, can be viewed as the repetition of the electrostatic discharge in the 1st domain at three locations.

In summary, compared with our previous work in 2019, the section "Cascaded-capacitor-breakdown model and experimental validation under ideal conditions"

offered a quantitative model with more real-world parameters. In other words, the model proposed in this section can be seen as an iteration and update of the findings and viewpoints in the previous work. Besides, a more important function of this section is providing a fundamental ideal model that can be extended to the comprehensive real-world model (the “cask model” that contains three discharge domains) demonstrated in the following sections.

As the analysis and study in this section are based on the same assumptions as previous research, we recognized a lack of explanation on how this section serves as a significant basis for the subsequent sections, which may confuse readers. To address this issue, we have added a description that connects the study in this section to the following sections, showing how the insights gained here deepen the understanding of the DC-TENG.

The revised part in the present manuscript is as follows:

We have made a revision of “The electric properties in the 1st, 2nd, and 3rd domains are similar, so FE and CCE, FE and TL can also be analyzed as capacitor models similar to the one in **Fig. 1c**. All possible discharge behaviors around DC-TENG can be regarded as the 1st discharge has been copied to the other two domains in some way.” in the second paragraph of “**Comprehensive “cask model” and optimization method of DC-TENG**”.

3. Too many abbreviations, making it hard to follow the context.

Response:

Thank you very much for your pertinent comment.

We are sorry for the confusion caused by the abbreviations in the original manuscript. We have carefully canceled some abbreviations and addressed this issue in the present manuscript with Red color.

The revised abbreviation of the present manuscript is shown in **Table R1**. The red font represents the reserved abbreviations in the present manuscript, while the black represents the canceled ones.

Table R1 Revised abbreviation

Abbreviation	Definition
5G	Fifth-generation
AI	Artificial intelligence
IoTs	Internet of Things
TENG	Triboelectric nanogenerator
DC-TENG	Direct-current triboelectric nanogenerator
RMS	Root-mean-square
CF	Crest factor
PVC	Polyvinyl chloride
PEEK	Polyether ether ketone
ETFE	Ethylene tetrafluoroethylene
FE	Friction electrode
CCE	Charge collecting electrode
TL	Triboelectric layer
BD	Breakdown
IL	Insulation layer
EIFE-DC-TENG	DC-TENG with end-isolated FE
SFE-DC-TENG	DC-TENG with split FEs
SCD	Surface charge density
EFI	Electric field intensity
FEM	Finite element method
HEGS	Hydroelectric energy generating system

4. As shown in Figure 6d, it seems the kapton IL attend the triboelectrification process. If it was case, then the output power enhancement will not due the isolation of the kapton, but the triboelectric property. Copper and kapton have different charge affinity, and the combination as shown in the figure, will definitely increase the output. That said, the conclusion is questionable.

Response:

Thank you very much for your careful review and professional comment.

We agree with you that copper and Kapton have different charge affinity, and the Kapton tape will contact the triboelectric layer (TL), which is a PVC film in our experiments. However, the contact electrification between Kapton and PVC has little influence on the performance of the direct-current triboelectric nanogenerator (DC-TENG). Kapton is a kind of dielectric material, and after a brief rubbing, the charge on the Kapton tape becomes saturated. Since the conductive electrode on the other side of the Kapton tape serves as a back electrode to bound charges², the discharge between Kapton and PVC rarely occurs. Hence, the saturated Kapton no longer participates in the contact electrification process, simply acting as an isolating tool with an ignorable influence on the electric field or electric output. This part can also be found in the present **Supplementary Note 17**.

An experiment has been conducted to verify the statement above. The data presented in **Fig. 6d** was used to calculate the charge densities collected by the DC-TENGs both without and with the 2nd domain blocked. Since copper and Kapton have different charge affinity and triboelectrification performance when rubbed against PVC film, we completely covered the surface of the friction electrode (FE) that faces TL and the edge of FE to investigate the role of Kapton in the triboelectrification process of DC-TENG operation. The results are shown in **Fig. R6a**. Compared with the charge density measured in the external circuit of DC-TENG with uncovered FE ($\sim 20 \mu\text{C m}^{-2}$ for the conventional configuration without 2nd domain blocked (**Fig. R6b**) and $\sim 65 \mu\text{C m}^{-2}$ for the 2nd domain blocked configuration (**Fig. R6c**), that of the DC-TENG with fully covered FE (**Fig. R6d**) can be negligible ($\sim 1.3 \mu\text{C m}^{-2}$). Hence, based on the experimental results, the triboelectrification contributed by Kapton type can be ignored and is not the main factor that increases the output performance of DC-TENG.

Figure R6 Triboelectric charge densities collected by the DC-TENG with different configurations. **a** Experimental results. **b** Conventional configuration of DC-TENG with the 2nd domain unblocked. **c** DC-TENG with the 2nd domain blocked. **d** DC-TENG with FE fully covered.

To avoid confusing readers, we have added the experimental results and necessary descriptions in the supplementary materials.

The revised part in the present supplementary materials is as follows:

We have added a supplementary figure as follows:

Supplementary Fig. 13 Triboelectric charge densities collected by the DC-TENG with FE fully covered configuration. Ten working cycles were measured for each trail. The error bar represents the standard deviation.

We have added “As shown in **Supplementary Fig. 13**, compared with the charge density measured in the external circuit of DC-TENG with uncovered FE (dozens of micro coulombs per square meter), that of the DC-TENG with covered FE configuration (insert of **Supplementary Fig. 13**) can be negligible ($\sim 1.3 \mu\text{C m}^{-2}$).”

At the end of “**Supplementary Note 17 Influence of the isolated layer on contact electrification process and output signals**”.

5. The "Cask model" that presented in Figure 6 need to be optimised. The 3rd BD, does not exist if FE and CCE are connected. Even the IL works, it will increase the speed of charge accumulation but not the level, because the discharge happen automatically when potential achieved to a certain level.

Response:

Thank you very much for your careful review and professional comment.

The occurrence of the 3rd breakdown is common in the direct-current triboelectric nanogenerator (DC-TENG) application scenario, and it holds significant value in the pursuit of high-performance DC-TENG. We provided a comprehensive discussion on the significance of the 3rd breakdown in our response to the first question.

For the comments on the isolation layer (abbreviated as IL in the original manuscript) function, whether it will increase the level depends mainly on the relative magnitude between the rate of triboelectric charge generation and the rate of charge release from the 1st domain. The rate of triboelectric charge generation is determined by the contact electrification process between the friction electrode (FE) and the triboelectric layer (TL). (As evidenced by **Supplementary Note 17** and **Supplementary Fig. 13**, the isolation layer has a negligible impact on the triboelectrification process, and therefore, does not significantly alter the rate of charge generation.) During the process of electrostatic breakdown in the 1st domain, the rate of charge release is not constant and contingents upon the potential difference between the two ends, relying on the condition of the external circuit condition.

Assuming complete isolation of the 2nd domain from electrostatic breakdown, when a DC-TENG with end-isolated FE starts to move, triboelectric charges are generated at a nearly constant rate while the rate of the 1st breakdown discharge is zero. As the charge accumulates, the breakdown threshold of the 1st domain is reached and the 1st breakdown is subsequently triggered. At this moment, the rate of charge flowing out from the 1st domain may be i) greater than, ii) equal to, or iii) less than the rate of

triboelectric charge generation. When the rate of charge released from the 1st electrostatic breakdown is greater than the rate of triboelectric charge generation, the accumulated charge will be released in a quite short time, after which it will accumulate again, resulting in pulse output signals. When the rate of the 1st breakdown is exactly equal to that of triboelectric charge generation, the entire process will reach a stable equilibrium.

When the rate of charge released from the 1st domain is less than the rate of triboelectric charge generation, the accumulated charge continues to increase, leading to an increase in the potential difference between FE/TL and the charge collecting electrode (CCE). A higher voltage forces the charge to be released from the 1st domain at a higher rate. The potential difference between FE/TL and CCE keeps increasing until the 3rd breakdown occurs or the rate of the charge flowing out from the 1st domain equals that of the triboelectric charge generation, which is a dynamic equilibrium in the real world. The rate of charge releasing from the 1st domain is subject to external load conditions, particularly when large resistive or capacitive loads are present or under open-circuit conditions, where the external loads seriously hinder reaching this dynamic equilibrium, and the rate of the charges releasing from the 1st domain may be diminished or even reduced to zero. Consequently, the charge is difficult/impossible to transfer and accumulate at both FE and CCE, generating a huge potential difference and triggering the 3rd breakdown. Hence, under this condition, not only does the speed of charge accumulation increase, but also the level increases.

A real-world example of this condition was observed in **Fig. R5**, where a DC-TENG with the 2nd domain blocked was fixed with FE facing upwards, and a transparent PVC film was adhered to the acrylic substrate as the TL. When subjected to a capacitive external load, the rate of charge released from the 1st domain is quite low. As TL rubbed against FE, the level increased to a specific magnitude, triggering the 3rd breakdown across the Kapton type and subsequently emitting bright light. Notably, even with Kapton tape adhered between FE and CCE, the discharge still occurs between the two electrodes, as evidenced by the bright discharge arc spanning

the tape. This phenomenon demonstrates the increased level of charge accumulation by introducing the isolation layer.

Figure R5 (Supplementary Fig. 10b, the left insert is Supplementary Fig. 10a) Evidence for the 3rd breakdown occurs on the DC-TENG with the 2nd domain blocked.

6. Some abbreviation is not defined, e.g. FE.

Response:

Thank you very much for your careful review and professional comment. We are very sorry for the confusion caused by abbreviations in the original manuscript. We have revised the abbreviations in the present manuscript and shown an abbreviation list in **Table R1**, in which we defined FE as friction electrode. The red font represents the reserved abbreviations in the present manuscript, while the black represents the canceled ones.

Table R1 Revised abbreviation

Abbreviation	Definition
5G	Fifth-generation
AI	Artificial intelligence
IoTs	Internet of Things

TENG	Triboelectric nanogenerator
DC-TENG	Direct-current triboelectric nanogenerator
RMS	Root-mean-square
CF	Crest factor
PVC	Polyvinyl chloride
PEEK	Polyether ether ketone
ETFE	Ethylene tetrafluoroethylene
FE	Friction electrode
CCE	Charge collecting electrode
TL	Triboelectric layer
BD	Breakdown
IL	Insulation layer
EIFE-DC-TENG	DC-TENG with end-isolated FE
SFE-DC-TENG	DC-TENG with split FEs
SCD	Surface charge density
EFI	Electric field intensity
FEM	Finite element method
HEGS	Hydroelectric energy generating system

Reviewer #2 (Remarks to the Author):

The manuscript NCOMMS-23-02049 “Discharge domains regulation and dynamic processes of direct-current triboelectric nanogenerator” proposes a new model to fully understand direct-current triboelectric nanogenerator (DC-TENG). This Reviewer does not have much concerns about the technical aspects and the electrical measurements, but some fundamental aspects regarding the mechanisms by which surfaces build-up charges (and the discussion section) need clarification. Also, the paper seems biased on TENGs and the real challenge here is to apply this model/results not only for DC-TENG, but also for tribology, triboelectrification and other electrostatic phenomena. I suggest that the authors address the following questions/concerns:

Response:

We highly appreciate the reviewer for carefully reviewing our work, and thank your positive comments on our research work. We have carefully considered all the suggestions and feedback provided, which has greatly enhanced our understanding of this field. We have taken your comments into account and added descriptions of the potential applications and significance of our findings to the revised manuscript.

The revised part in the present manuscript is as follows:

We have added “Furthermore, the models and results in this paper may also be applied to tribology, triboelectrification, and other electrostatic phenomena. In a word, ...” in **Discussion** section.

1. The generation of (only) opposite charges on FE and TL upon friction (or contact) is not supported by many important papers. In fact, bipolar, multipolar and mosaics of charges on tribo surfaces have been imaged and identified at both micro and macro scales. Thus, the first paragraph in the results section and Figure 1/Figure 4/Figure 5 are not accurate since the flow of charges and the net charge are different from charge domains. If this is the case, authors must provide additional results. For example, this claim could be supported by electrostatic mappings of FE and TL done with macroscopic Kelvin probes. Charge domains could also be revealed with a charged tonner (e.g. Xerox® Cyan Developer – powder iGen3, 5R706) or other means. Kelvin

probe force microscopy can also help the discussion, but only as a complementary tool since charge domains are macroscopically distributed.

Response:

Thank you very much for your careful review and professional comment.

We are very sorry for the inaccurate statement in the first paragraph and Figure 1/Figure 4/Figure 5. In this work, we focus on the flow of charges and the net charge since this manuscript pays more attention to the output electrical signals. In experiments presented in this manuscript, after the contact action between the dielectric layer (polyether ether ketone (PEEK), polyvinyl chloride (PVC), and ethylene tetrafluoroethylene (ETFE)) and the friction electrode (copper), the tribo surface of the dielectric layer is net negatively charged though there possibly be positive charges on it, while that of the friction electrode (FE) is net positively charged.

We do not exclude the possibility or deny that these possible effects (bipolar, multipolar, and mosaics of charges) can form on the two tribo surfaces. However, we were unable to detect signals that could confirm the existence of these phenomena, and the possible reasons are shown below:

i) We used a metal-dielectric material-pair to perform contact electrification instead of dielectric-dielectric which is usually employed when studying the bipolar, multipolar, and mosaics patterns. Since the triboelectric charges in metal are redistributed and are influenced by the electric field around, it is challenging to identify these phenomena on the tribo surface of the metal.

ii) The motion in our manuscript is sliding mode, different from the contact-separation or peeling modes used in previous research. From earlier studies in the field of sliding mode triboelectric nanogenerator (S-TENG), the inducted signals do not confirm the existence of these possible effects.

iii) In our manuscript, electrostatic breakdowns occur between the material-pairs of metal-dielectric and metal-metal, instead of dielectric-dielectric pairs. The distribution of charges inside a metal is influenced by the entire electric field in the space, resulting in the breakdowns more prone to show the characteristics of the net charge on the entire surface, rather than the local charge domains.

In summary, possibly due to different materials' electrical characteristics and experimental configurations in the contact electrification and electrostatic breakdown processes, we were not able to catch the signals of bipolar, multipolar, or mosaics of charges. Based on the main focus of our manuscript, the resultant electrical performance of DC-TENG, we pay more attention to the net charge, which is the superposition of all charges present on the entire surface. From another point of view, bipolar, multipolar, and mosaics of charges can be regarded as a form of unevenness during electrification. The unevenness of these phenomena involves not only the magnitude but also the polarity of charges. Such unevenness can introduce a degree of randomness to the rate of charge generation, which can be modeled by random factor α in our manuscript. In practice, regardless of how the local bipolar, multipolar, and mosaic patterns perform, the overall equivalent charge remains relatively stable and is the main focus of our work.

To describe this contact electrification process more comprehensively and accurately, we have cited more related articles and added a note in supplementary materials.

The revised part in the present manuscript is as follows:

We have made a revision of “Friction between FE and the TL generates net opposite charges on both contact surfaces due to contact electrification (**Fig. 1b(i)**). (To avoid possible confusion and enhance the readability, the term "net charge" will be referred to as simply "charge" in the following text^{35,36}. **Supplementary Note 2** shows more analysis and details.) As FE slides on TL, ...” in the first paragraph of **Results** section.

We have added “

35. H. Baytekin, et al. The mosaic of surface charge in contact electrification. *Science* **333**, 308-312 (2011).
36. Y. I. Sobolev, W. Adamkiewicz, M. Siek & B. A. Grzybowski Charge mosaics on contact-electrified dielectrics result from polarity-inverting discharges. *Nat. Phys.* **18**, 1347-1355 (2022).” In the **Reference** section of the manuscript.

The revised part in the present supplementary materials is as follows:

“**Supplementary Note 2 Note on the surface charge**

As some previous work declared, the bipolar, multipolar, and mosaics of charges on tribo surfaces can occur in micro and macro scales, especially for the contact between two dielectric layers.^{2,3} However, it is worth noting that these effects can be simplified to an equivalent charge, specifically the net charge, when applying contact electrification to TENG. This work focuses more on the output electrical performance of TENG, and the measured output signal is the result of the electric field generated by the superposition of all charges present in the entire surface. In practice, regardless of how the local bipolar, multipolar, and mosaic patterns perform, the overall equivalent charge remains relatively stable. Hence, under this condition, for the sake of simplicity, the terms such as "charge", "surface charge density", "residual charge", and other similar expressions associated with electrostatic charges generated by contact electrification in this work refer to a simplified expression of "net charge", "net surface charge density", "net residual charge", and so forth.”

We have added “

2. H. Baytekin, et al. The mosaic of surface charge in contact electrification. *Science* **333**, 308-312 (2011).
3. Y. I. Sobolev, W. Adamkiewicz, M. Siek & B. A. Grzybowski Charge mosaics on contact-electrified dielectrics result from polarity-inverting discharges. *Nat. Phys.* **18**, 1347-1355 (2022).” in the **Reference** section of the supplementary materials.

2. Some of the work reported here is similar to the work reported on DOI: 10.1002/anie.201806658 to indicate the effects caused by distant substrates on the outcome of contact electrification. Also, a recent paper (DOI: 10.1038/s41567-022-01714-9) deals with electrostatic charging during peeling, showing that charge domains are formed not due properties of the contacting materials but due to electrostatic discharges between the separating surfaces. In fact, sparks not only dissipate the surface charges but invert their polarities at the ends of the spark. Thus, electric current oscillations reported in the manuscript could have the same origin as those reported in the mentioned papers.

Response:

Thank you very much for your informative comment. As the reviewer’s suggestion,

we restudied the two works carefully.

The first work³ (DOI: 10.1002/anie.201806658) revealed that after the contact electrification process between the two dielectric materials, the distribution of the surface charges on the polymer is greatly affected by the arc discharge that occurs on the distant conductive substrate. Instead of participating in the contact electrification process, the conductive substrate influenced the electric field by electrostatic induction and electrostatic breakdowns. However, the friction electrode (FE) in our work directly participated in the contact electrification process. It was involved in both contact electrification and electrostatic breakdown, which makes it difficult to observe and character the electrostatic breakdown. Thus, the proposed configuration of direct-current triboelectric nanogenerator (DC-TENG) with split FEs in **Fig. 5a** was valuable, as it divided the two physical processes and allowed us to study them separately. Moreover, this work³ focused on the resultant surface charge density caused by contact electrification (dielectric-dielectric) and discharge (distant metal), while our work can be regarded as a kind of utilization of the resultant outcome from contact electrification (dielectric-metal) and electrostatic breakdown (dielectric-metal) with a different mechanical configuration, more attention put on the net output electrical signals.

The second work⁴ (DOI: 10.1038/s41567-022-01714-9) proposed theoretical models and experimental results to explain the formation of the mosaic patterns on the tribo surfaces. Based on peeling experimental configuration settings, they suggested that the mosaic pattern is formed due to the electrostatic discharges between the separating surfaces, which may also cause an inversion of the polarities to the surface charges. In this study⁴, an intrinsic model was developed based on Paschen's law to simulate the charge density of the contact-charge surfaces along the x-axis. Our manuscript presents a numerical simulation model under ideal conditions, in which the output signals dynamically change over time. Based on this numerical model and the corresponding experiments, electric current oscillations reported in the manuscript have two possible origins:

i) Randomness during the contact electrification. Slight fluctuations in the simulated current results can be attributed to randomness (contact efficiency, uneven

distribution of charges, non-uniform motion, and other possible factors) encountered in the numerical model, as discussed in **Supplementary Fig. 3** and **Supplementary Note 6**.

ii) Electrostatic breakdown threshold. The intense oscillation signals, or pulsed signals, corresponding to the discharge in the 1st domain under ideal condition (ideal condition: except for a small number of residual charges left on triboelectric layer (TL) surface, charges generated by the friction between FE and TL are released by electrostatic breakdown at charge collecting electrode (CCE)).

Whether electric current oscillations reported in our manuscript have the same origin as those reported in the previous papers needs more theoretical analysis and experimental verification. On one hand, the motion mode (contact-separation or sliding) and the friction materials (dielectric-dielectric material pair or dielectric-metal material pair) in our manuscript are different from those in the previous work; on the other hand, the electric current shown in our manuscript is the output of DC-TENG, which is an energy converter designed based on contact electrification (between FE and TL) and electrostatic discharge (between CCE and TL), different from the oscillating charge density measured in the previous work mentioned above.

To help potential readers understand the contact electrification and electrostatic discharge processes more comprehensively, we have cited these two articles and made more discussion.

The revised part in the present manuscript is as follows:

We have made a revision of “Thus, all the potential discharge locations are defined into three categories due to their characters and influence on the output performance. **With the systematic analysis on discharge domains of DC-TENG, more fundamental theories about contact electrification and electrostatic discharge^{36, 39} can be utilized to understand and further improve its performance, which previously relied on empirical observations and phenomena. Furthermore, the models and results in this paper may also be applied to tribology, triboelectrification, and other electrostatic phenomena. In a word, the “cask model” introduced in this work not only points out the direction for substantially increasing and regulating outputs, opening a new chapter of DC-TENG**

dynamic outputs, but also provides insights into the underlying physical mechanisms, ultimately leading to a more comprehensive understanding of this fascinating field.” in the **Discussion** section.

We have added “

36. Y. I. Sobolev, W. Adamkiewicz, M. Siek & B. A. Grzybowski Charge mosaics on contact-electrified dielectrics result from polarity-inverting discharges. *Nat. Phys.* **18**, 1347-1355 (2022).

...

39. M. Siek, W. Adamkiewicz, Y. I. Sobolev & B. A. Grzybowski The influence of distant substrates on the outcome of contact electrification. *Angew. Chem. Int. Ed.* **57**, 15379-15383 (2018).” In the **Reference** section of the manuscript.

3. How the surface charge density is effectively calculated? If the authors are measuring the transferred charge using the Coulomb function (and/or the current function through Keithley 6514 triax cable high input) this is not surface charge density. For example, charges must flow through the junctions of contact at the interface developed due to asperities of both contacting surfaces. In other words, electric charge is being transferred at the real area of contact, usually less than 1% of the apparent contact area. Even thermal fluctuations at the metal-dielectric interface during sliding contact can affect the measuring charges. Again, electrostatic mappings would bring rich information to this manuscript.

Response:

Thank you very much for your careful review and professional guidance.

We fully agree with the comment that the real area of contact is significantly smaller than the apparent contact area when we zoom into the contact junction of the friction electrode (FE) and the triboelectric layer (TL). In this manuscript, the term “surface charge density” refers to the transferred charge, measured by Keithley 6514, divided by the area swept over by the sliding motion. We are sorry for the lack of a clearer definition of this important term in the original manuscript.

To comprehensively investigate the contact electrification and electrostatic discharge processes, it is essential to consider the real area of contact. However, accurately measuring it remains a long-standing challenge as no mature technology has

been reported yet.^{5, 6} Our work primarily focuses on improving the output electrical performance of DC-TENG through theoretical analysis and feasible experiments. In this context, DC-TENG can be viewed as an application that harnesses the net charge density. For our practical engineering purposes, practicality, simplicity, and applicability are more critical than microscopic accuracy. Therefore, in engineering applications, it is sufficient to use surface charge density based on the apparent area. Moreover, this surface charge density based on the apparent area plays a crucial role in the overall model proposed in this manuscript, particularly in the first section where we analyze the three main aspects that influence the output performance (material parameters, structural parameters, motion parameters). It facilitates cooperation with the width of FE to simulate the output performance of the DC-TENG. Therefore, it is sufficient for our proposed model.

Thank you again for your valuable comment. It pointed out a promising direction for improving the output performance of DC-TENG, which is optimizing the contact efficiency and increasing the real area of contact. While using apparent area is sufficient for our current work, we acknowledge the importance of a deeper understanding of the contact electrification and electrostatic discharge processes, and we will continue to investigate them from a microscopic view in future work. We greatly appreciate your professional and insightful comment.

To avoid possible confusion to readers, we have emphasized the definition of the term “surface charge density” in the present manuscript.

The revised part in the present manuscript is as follows:

We have made a revision of “The rate of charge flowing into C_{CCE-TL} can be modeled as the product of the **surface charge density** collected at CCE σ_{CCE} (transferred charge measured by an electrometer divided by the area swept over by **relative sliding**), the width of CCE w_{CCE} , and relative sliding speed $v_{sliding}$ (Supplementary Note 5).” in “**Cascaded-capacitor-breakdown model and experimental validation under ideal conditions**”.

Some minor points:

1) The effect of rougher TL surfaces should be discussed, as contact junctions are important to contact electrification and may also play an important role in the cascaded-capacitor-breakdown model.

Response:

Thank you very much for the invaluable suggestion.

We strongly agree with the statement that contact junctions are important to contact electrification and the cascaded-capacitor-breakdown model. The roughness of the triboelectric layer (TL) can affect contact electrification performance by altering the real area of contact and contact pressure. In the cascaded-capacitor-breakdown model, a random factor α was employed to take the conditions mentioned above into consideration. Besides the roughness, α also serves as a comprehensive term to account for the unevenness introduced by materials uniformity and other properties in the contact electrification process. It conforms to the normal distribution. With this random factor, the rate of charge flowing into the capacitor is not a constant, but a value with a kind of randomness (**Supplementary Fig. 3**), which is consistent with the practical conditions. This factor was calibrated based on pre-experimental results first and then utilized to numerically simulate the ideal dynamic output of DC-TENG. The cascaded-capacitor-breakdown model thus offers a comprehensive representation of the inconsistency caused by material uniformity, roughness of the two tribo surfaces, and other related factors.

Moreover, in this cascaded-capacitor-breakdown model, the roughness also influences the electrostatic breakdown phenomena. During the triboelectrification process, a non-uniform charge distribution is formed on the TL surface, affecting the distribution and orientation of the charge domains, which in turn affects the electric field intensity and can result in a further non-uniform electrostatic breakdown process. Additionally, the roughness of the TL can make the distance between the TL surface and the bottom surface of the charge collecting electrode (CCE) inconstant. According to Paschen's law, different air gaps lead to different breakdown characteristics. Therefore, to reflect real-world conditions, the cascaded-capacitor-breakdown model

occupies multiple secondary capacitor units. The entire focusing region between CCE and TL is divided into multiple categories based on a statistical analysis of distance in microscopic view, and the parameters involved in this cascaded structure algorithm were determined with numerical methods.

In one word, the roughness influences the performance of DC-TENG and the cascaded-capacitor-breakdown model considered it on both contact electrification and electrostatic breakdown phenomena with parameters calibrated numerically.

To make it more comprehensive, we have added some discussion about the roughness and revised related content in the present supplementary materials.

The revised part in the present supplementary materials is as follows:

We have made a revision of “

Supplementary Note 6 The random factor in the capacitor breakdown model

The roughness of TL can affect contact electrification performance by altering the real area of contact and contact pressure. Considering the charge generation side, a random factor α is employed to take practically inconsistent conditions into account, such as the uniformity of materials, the roughness of two contacting surfaces, and contact efficiency. It conforms to the normal distribution (**Supplementary Fig. 3a**). With this random factor, the rate of charge flowing into the capacitor is not a constant (**Supplementary Fig. 3b**), but a value with a kind of randomness (**Supplementary Fig. 3c**), which is consistent with the practical conditions.” in “**Supplementary Note 6 The random factor in the capacitor breakdown model**”.

We have made a revision of “

Supplementary Note 7 The division strategy for secondary capacitors in the cascaded-capacitor-breakdown model

In this cascaded-capacitor-breakdown model, the roughness also influences the electrostatic breakdown phenomena. The roughness of the TL can make the distance between the TL surface and the bottom surface of the CCE inconstant. According to Paschen’s law, different air gaps lead to different breakdown characteristics. Therefore, to reflect real-world conditions, the cascaded-capacitor-breakdown model occupies

multiple secondary capacitor units. When it comes to the discharge process, the capacitor composed of CCE and TL is further divided into several secondary capacitor units according to the real distance between CCE and TL, since the two facing surfaces of CCE and TL are not perfectly flat from microscopic view due to the practical factors, such as material properties and fabrication process (**Supplementary Fig. 4a**). To be more specific, by dividing the whole focusing region between CCE and TL into two categories due to the statistical analysis in the microscopic view, the area with the distance of d_{\min} to $\frac{(d_{\max}-d_{\min})}{2}$ is divided into one category, and that of $\frac{(d_{\max}-d_{\min})}{2}$ to d_{\max} is divided into another category (**Supplementary Fig. 4b**). Similarly, **Supplementary Fig. 4c** and **Supplementary Fig. 4d** illustrate the situation when more categories are classified.” in “**Supplementary Note 7 The division strategy for secondary capacitors in the cascaded-capacitor-breakdown model**”.

2) A triboplasma is frequently generated under sliding contact conditions. Thus, triboemission and triboluminescence processes could be occurring while electrical charges ionize the surrounding air, which would have the same outcome registered in Figure 4. Thus, the claim that corona discharge is occurring at the interface needs further verification. This could be confirmed by conducting the experiments in both neon gas atmosphere and under high vacuum ($< 10^{-7}$ bar).

Response:

Thank you very much for your careful review and professional comment.

Corona discharge is an electrical discharge occurs at locations where the strength of the electric field around a conductor exceeds the dielectric strength of the air. It represents a local region where the air has undergone electrical breakdown and become conductive, allowing charge to continuously leak off the conductor into the air. Corona process applications emphasize one of two aspects of the discharge: the ions produced or the energetic electronic producing the plasma⁷.

Triboplasma refers to the plasma generated at the sliding contact interface. In 2002, Nakayama reported a discovery of triboplasma generated in the microscopic gap⁸,

which confirmed the hypothesis presented in 1965⁹. The evidence on the emission of electrons, ions, and photons in ambient air^{10, 11} were also provided. Current research showed that triboplasma, together with triboemission and triboluminescence processes, also follows Paschen's law and can occur under atmospheric pressure¹² as corona discharge does. And as the reviewer's comment, it would have the same outcome registered in **Fig. 4**.

In some way, the corona discharge mentioned in our manuscript has a similar meaning as triboplasma, since the main material involved in the corona discharge is plasma and the high-voltage condition is reached by triboelectrification. "Corona discharge" emphasizes the electrical discharge type, while "triboplasma" points out the trigger method is friction and the resultant matter is plasma.

To verify the discharge type, we conducted an experiment with a home-made high vacuum system to keep the working environment of $3\text{-}5\times 10^{-5}$ Pa at room temperature. A mechanical pump combined with a molecule pump was used for realizing the ultimate high vacuum condition. The same DC-TENG device that experienced continuous breakdown in atmospheric pressure (**Fig. R7a**), in the high vacuum condition, outputs a sudden net charge increase after a period of sliding (**Fig. R7b**), which is the triboelectrification process between the friction electrode (FE) and the triboelectric layer (TL). We hypothesize that, instead of corona discharge, another type of electrostatic breakdown occurs in the high vacuum condition (most likely the field emission), which limits the surface charge density. This type of breakdown requires a higher surface charge density compared to the breakdown threshold in the air. Thus, the charge density on the surface of TL needs to be accumulated for a while and then breakdown at an instant.

In summary, while we cannot exclude the possibility of other types of electrostatic breakdown during the operation of DC-TENG, corona discharge may be the main type since the breakdown threshold suggested in the vacuum condition is much higher than that of corona discharge in the air.

Figure R7 Output of conventional DC-TENG. a Under atmospheric pressure condition. **b** Under high vacuum condition.

3) With the experimental conditions mentioned before, authors can provide Paschen plots. In fact, recent works have shown that for vertical distances in the micrometer range, the breakdown voltage shows a significant reduction. Moreover, it is very unusual that the Paschen law is not even mentioned in a paper claiming that the results are explained through an electrostatic breakdown model.

Response:

Thank you very much for your careful review and professional comment.

We are very sorry for the lacking of an elaborate explanation of Paschen's law in the original manuscript. Here, we provided supplementary information on the detailed analysis process.

In our manuscript, the working condition was in the air of 1 atmosphere, so we calculated the breakdown voltage based on Paschen's law and plotted it in **Fig. R8**. The equation is shown in **Equation R1**, and the values of the parameters in this equation are presented in **Table R2**.

$$V_b = \frac{Bpd}{\ln(Apd) - \ln[\ln(1 + \frac{1}{\gamma})]} \quad (\text{R1})$$

Table R2 Paschen's constants for air

A	B	γ
(m ⁻¹ Pa ⁻¹)	(V m ⁻¹ Pa ⁻¹)	(×10 ⁻³)
10.95	273.8	8.136

Figure R8 Paschen's curve: Gas is air at a pressure of 1 atmosphere.

While researchers generally accept the validity of the Paschen's law for distances above 5-10 microns (i.e., the right side of the curve in **Fig. R8**), there is ongoing debate regarding its applicability for smaller air gaps (i.e., the left side of the curve in **Fig. R8**)¹³⁻¹⁵, recently. In our work, we have to utilize the breakdown voltage at distances ranging from ~ 0 (the contact point of the friction electrode (FE) and the triboelectric layer (TL)) to millimeter scale (the distance between FE and charge collecting electrode (CCE)), which falls within the controversial region where the accuracy of Paschen's law is debated. We derived the breakdown threshold of the electric field intensity from the calculated breakdown voltage, which varied from 3 MV m^{-1} to 7 MV m^{-1} for the right side of Paschen's curve. The DC-TENG device itself presents many non-ideal factors that further complicate the analysis of its electrical properties. For instance, the electric field around the DC-TENG is not uniform as assumed in Paschen's law, which is probably due to the mechanical configuration, materials selections, and surface condition issues. To simplify our model and emphasize the main focus of this work, we

used a breakdown voltage magnitude that many researchers have employed, which is 3 MV m^{-1} ^{14, 16-18}.

For the left side of the curve, some works stated that in addition to the Townsend avalanche of gaseous ions, other effects such as field emission of electrons and tunneling electrons have been proposed as potential factors, and they offered a modified Paschen's curve with a slope of $60\text{-}80 \text{ MV m}^{-1}$ ¹³⁻¹⁵. Whether considering the original Paschen's curve or the modified one, the breakdown electric field intensity exceeds 3 MV m^{-1} by more than an order of magnitude, which does not align with real-world conditions. For instance, the electric field intensity threshold for triggering the field emission is $\sim 75 \text{ MV m}^{-1}$ ¹⁴, requiring ultra-high surface charge density of $\sim 1330 \mu\text{C m}^{-2}$, which is a strict condition to achieve during the sliding motion in the air with 1 atmosphere. Hence, 3 MV m^{-1} can be a proper magnitude of breakdown threshold.

To avoid possible confusion, we have added the discussion about Paschen's law in the present supplementary materials and referred to it in the present manuscript.

The revised part in the present manuscript is as follows:

We have made a revision of “As charges continuously flow into the capacitor $C_{\text{CCE-TL}}$, the electric field intensity between CCE and TL keeps increasing until the breakdown threshold, which is 3 MV m^{-1} in the air, is reached. (Discussion about the breakdown threshold is shown in **Supplementary Fig. 1** and **Supplementary Note 3**) When the electric field intensity exceeds the threshold, ...” in the second paragraph of “**Cascaded-capacitor-breakdown model and experimental validation under ideal conditions**”.

The revised part in the present supplementary materials is as follows:

We have added a supplementary figure as follows:

Supplementary Fig. 1 Paschen's curve: Gas is air at a pressure of 1 atmosphere.

We have added a supplementary note as follows:

“Supplementary Note 3 Determination of the threshold of the electric field intensity base on Paschen's law

Paschen's law for gas breakdown gives an empirical relation between the breakdown voltage and the product of the gas pressure and the distance between the two electrodes involved in the breakdown process. The question is shown in **Equation S1**.

$$V_b = \frac{Bpd}{\ln(Apd) - \ln\left[\ln\left(1 + \frac{1}{\gamma}\right)\right]} \quad (\text{S1})$$

where V_b is the breakdown voltage; p is the gas pressure; d is the distance; A and B are constants determined by the pressure and composition of the gas; and γ is the secondary electron emission coefficient.

In our manuscript, the working condition was in the air of 1 atmosphere, so we plotted the Paschen's curve in **Supplementary Fig. 1** with $A = 10.95 \text{ m}^{-1} \text{ Pa}^{-1}$, $B = 273.8 \text{ V m}^{-1} \text{ Pa}^{-1}$, and $\gamma = 8.136 \times 10^{-3}$.

While researchers generally accept the validity of the Paschen's law for distances above 5-10 microns (i.e., the right side of the curve in **Supplementary Fig. 1**), there is ongoing debate regarding its applicability for smaller air gaps (i.e., the left side of the curve in **Supplementary Fig. 1**)⁴⁻⁶, recently. In our work, we have to utilize the breakdown voltage at distances ranging from ~ 0 (the contact point of the friction electrode (FE) and the triboelectric layer (TL)) to millimeter scale (the distance between FE and charge collecting electrode (CCE)), which falls within the controversial region where the accuracy of Paschen's law is debated. We derived the breakdown threshold of the electric field intensity from the calculated breakdown voltage, which varied from 3 MV m^{-1} to 7 MV m^{-1} for the right side of Paschen's curve. The DC-TENG device itself presents many non-ideal factors that further complicate the analysis of its electrical properties. For instance, the electric field around the DC-TENG is not uniform as assumed in Paschen's law, which is probably due to the mechanical configuration, materials selections, and electromagnetic environment issues. To simplify our model and emphasize the main focus of this work, we used a breakdown voltage magnitude that many researchers have employed, which is 3 MV m^{-1} ^{5, 7-9}.

For the left side of the curve, some works stated that in addition to the Townsend avalanche of gaseous ions, other effects such as field emission of electrons and tunneling electrons have been proposed as potential factors, and they offered a modified Paschen's curve with a slope of $60\text{-}80 \text{ MV m}^{-1}$ ⁴⁻⁶. Whether considering the original Paschen's curve or the modified one, the breakdown electric field intensity exceeds 3 MV m^{-1} by more than an order of magnitude, which does not align with real-world conditions. For instance, the electric field intensity threshold for triggering the field emission is $\sim 75 \text{ MV m}^{-1}$ ⁵, requiring ultra-high surface charge density of $\sim 1330 \text{ } \mu\text{C m}^{-2}$, which is a strict condition to achieve in the air with 1 atmosphere. Hence, 3 MV m^{-1} can be a proper magnitude of breakdown threshold."

We have added "

4. D. B. Go & D. A. Pohlman A mathematical model of the modified Paschen's curve for breakdown in microscale gaps. *J. Appl. Phys.* **107**, 103303 (2010).
5. A. J. Wallash & L. Levit Electrical breakdown and ESD phenomena for devices with nanometer-to-micron gaps. *Proc. SPIE* **4980**, 87-96 (2003).

6. A. Peschot, N. Bonifaci, O. Lesaint, C. Valadares & C. Poulain Deviations from the Paschen's law at short gap distances from 100 nm to 10 μm in air and nitrogen. *Appl. Phys. Lett.* **105**, 123109 (2014).
7. D. Liu, et al. A constant current triboelectric nanogenerator arising from electrostatic breakdown. *Sci. Adv.* **5**, eaav6437 (2019).
8. D. Liu, L. Zhou, Z. L. Wang & J. Wang Triboelectric nanogenerator: From alternating current to direct current. *iScience* **24**, 102018 (2021).
9. Z. Zhao, D. Liu, Y. Li, Z. L. Wang & J. Wang Direct-current triboelectric nanogenerator based on electrostatic breakdown effect. *Nano Energy* 107745 (2022).” in the **Reference** section of the supplementary materials.

4) What was the relative humidity and temperature conditions? Adsorbed water plays a key role on both charging and discharging phenomena. Actually, water that is frequently described as playing a passive role on triboelectrification (increasing surface conductivity), but many papers have been describing its active role on charging and induction. Authors must discuss this within the text.

Response:

Thank you very much for your careful review and professional comment. We are sorry that we did not clearly describe the relative humidity and temperature conditions of our experiments.

We conducted all experiments in a controlled laboratory environment, maintaining a temperature range of 293.15-298.15 K and a relative humidity level of 10%-20%.

As for the influence of water on the charging and discharging processes, we totally agree with you that it plays a crucial role. Here we discussed the role of water in the charging and discharging processes separately, and also analyzed its function in our DC-TENG, which combines both contact electrification and electrostatic discharge phenomena.

The role of water in the charging process:

Previous papers generally conducted triboelectrification experiments under two conditions: contact-separation mode and sliding mode. For experiments with contact-separation mode experimental settings, the results generally turn out that water plays a negative role in triboelectrification. The possible reasons are as follows: i) The water molecules adsorbed on the friction surface in a high humidity environment will

dissipate the triboelectric charge through the conductive path formed by water¹⁹⁻²³; ii) Water molecules adsorbed between the metal/polymer interfaces, due to the existence of hydrogen bonds, replacing electron donors and reducing electron transfer, decreasing charge transferred under humid conditions²⁴. For experiments involving sliding mode (and free-standing mode), the amount of triboelectric charge initially increases with increasing relative humidity and then decreases beyond a certain value, indicating an optimal working humidity^{23, 25}. The possible explanation is that in a low-humidity environment, as the relative humidity increases, the dispersed water bridge formed between the friction materials, filling the grooves on the tribo surfaces under microscopic view, providing more channels for transferred electrons²⁵ or ions²⁶, resulting in higher output performance. When the relative humidity is further increased, a continuous water layer forms on the surface, resulting in a decrease in the number of transferred electrons²⁵.

Besides, the effect of humidity on triboelectrification also depends on the materials occupied. In an early research²⁷, as the relative humidity increases from 40% to 80%, the output voltage of a TENG based on a hydrophilic polyamide film decays logarithmically, while the voltage changes of a TENG based on a hydrophobic polytetrafluoroethylene film is negligible. Consequently, the changes in output performance under humid conditions are possibly influenced by the surface free energy and hydrophobicity of the polymer. As the humidity increases, water molecules adsorb onto the surface of hydrophilic materials, creating a conductive water layer that dissipates surface charge. Conversely, hydrophobic surfaces hinder water molecule adsorption and effectively inhibit water molecule spreading on the surface, maintaining stable output performance in high-humidity environments.^{21, 28, 29} Additionally, larger surface contact angles can suppress surface discharge effects in high-humidity environments.²³ Some water-adsorbing materials can immobilize water molecules to participate in contact electrification, such as polyvinyl alcohol^{26, 30-32}, starch molecules³³, and cellulose³⁴, thereby reducing the negative impact of humidity on triboelectrification, or even increasing the output performance of TENG²². The electrification and surface charge retention ability of dielectric materials in high-

humidity environments are determined not only by their hydrophobicity but also by their inherent contact electrification properties. In addition to hydrophobicity, the surface chemical properties related to functional groups and surface topography characterized by roughness may also affect triboelectrification performance at different humidity levels³⁵.

The role of water in the discharging process:

For discharging process, the humidity is also one of the important environmental factors, which may alter the breakdown characteristics of the gas. As the relative humidity increases from 30% to 90%, the breakdown voltage of the rod-plate air gap noticeably decreases, from 843.3 V to 796.7 V³⁶. Similar to the charging process, materials also play a significant role in the discharge process. Compared with hydrophobic dielectric materials, water molecules are more easily adsorbed on the surface of hydrophilic materials, which may influence the electrostatic field in the gap.

The role of water in DC-TENG:

Since DC-TENG arises from both triboelectrification and electrostatic discharge, the output performance is influenced by both physical processes. As our earlier work showed, the utilization rate of triboelectric charges increases in a high humidity environment (tested with PTFE triboelectric layers (TL) and copper friction electrode (FE)), indicating that more charges participate in the electrostatic breakdown process³⁶. This is because a high-humidity environment not only promotes the electrification effect of the hydrophobic materials to generate more triboelectric charges, but also reduces the gap breakdown voltage, thus facilitating the electrostatic breakdown process. As a result, the output performance is improved in high-humidity environments.

Conclusion:

In summary, the presence of adsorbed water is crucial for both the charging and discharging processes in TENGs. Its role in the triboelectrification process depends on the motion mode (contact-separation mode, sliding or free-standing mode), materials (chemical and physical properties characterized by contact angle, functional group, roughness, and inherent triboelectrification ability), and other operational parameters.

Generally, in contact-separation mode, humidity has a negative effect on triboelectrification, while the influence of the humidity on the sliding mode TENG is less or even positive. The use of hydrophilic materials as the triboelectric layer may lead to a negative effect of high humidity on the contact electrification effect, whereas the use of hydrophobic materials may result in an ignorable or positive effect. In the discharging process, higher humidity leads to a lower breakdown voltage, making it easier to trigger electrostatic breakdown. Integrated the two physical processes of triboelectrification and electrostatic breakdown into the DC-TENG, the output performance of the DC-TENG is improved in a high-humidity environment, particularly when the hydrophobic materials are used as the TL.

To avoid possible confusion, we have added discussion about the influence of water on both triboelectrification and electrostatic breakdown, and revised the description involving experiments with more details about the working environment in the present manuscript.

The revised part in the present manuscript is as follows:

We have made a revision of "... a cascaded-capacitor-breakdown model was built for the ideal condition and confirmed with experiments, revealing that under a relatively constant environmental condition (**Supplementary Note 20**), the main factors that influence the output performance of DC-TENGs are ..." in **Discussion** section.

We have added description of the working environment of "**Environmental setting**. To minimize the influence of environmental factors on the experimental results, the temperature and relative humidity were carefully controlled in a laboratory environment. Unless otherwise specified, all experiments were conducted at a temperature range of 293.15-298.15 K and a relative humidity level of 10%-20%." in **Methods** section.

The revised part in the present supplementary materials is as follows:

We have added a supplementary note as follows:

"Supplementary Note 20 Influence of environmental conditions

For the sake of controlling variables and simplifying the theoretical model and experiments to a certain extent, consistent environmental parameters were employed throughout this work. However, it is worth noting that in practical applications, both temperature and relative humidity can impact the operation of DC-TENG. The presence of adsorbed water on the tribo surfaces is crucial for both the charging and discharging processes¹⁴. Its role in the triboelectrification process depends on the motion mode (contact-separation mode, sliding or free-standing mode), materials (chemical and physical properties characterized by the contact angle, functional group, roughness, and inherent triboelectrification ability), and other operational parameters. For discharging process, humidity may alter the breakdown characteristics of the gas, resulting in a noticeable decrease in the breakdown voltage as humidity increases¹⁵. As the DC-TENG device arises from both triboelectrification and electrostatic discharge, its output performance is influenced by both physical processes. Some research suggested that the output performance of DC-TENG is improved in high-humidity environments, as a high-humidity environment not only promotes the electrification effect of the hydrophobic materials to generate more triboelectric charges but also reduces the gap breakdown voltage, facilitating the electrostatic breakdown process. Nevertheless, the complex influence of humidity on the two physical processes is still controversial, and its influence on the output performance of DC-TENG needs more comprehensive study.”

We have added “

14. L. Liu, et al. A high humidity-resistive triboelectric nanogenerator via coupling of dielectric material selection and surface-charge engineering. *J. Mater. Chem. A* **9**, 21357-21365 (2021).
15. L. Liu, et al. Achieving ultrahigh effective surface charge density of direct-current triboelectric nanogenerator in high humidity. *Small* **18**, 2201402 (2022).” in the **Reference** section of the supplementary materials.

5) The paper is full of acronyms and any reader gets really tired after the first three pages. Authors must bring another solution for this issue. Also, the FE acronym is not explained through the text. From Figure 1, I am assuming that “FE” is the “Friction Electrode”.

Response:

We are very sorry for the confusion caused by abbreviations in the original manuscript. We have carefully canceled some abbreviations and addressed this issue in the present manuscript with Red color. We have added the definition of FE, which is friction electrode.

The revised abbreviation of the present manuscript is shown in **Table R1**. The red font represents the reserved abbreviations in the present manuscript, while the black represents the canceled ones.

Table R1 Revised abbreviation

Abbreviation	Definition
5G	Fifth-generation
AI	Artificial intelligence
IoTs	Internet of Things
TENG	Triboelectric nanogenerator
DC-TENG	Direct-current triboelectric nanogenerator
RMS	Root-mean-square
CF	Crest factor
PVC	Polyvinyl chloride
PEEK	Polyether ether ketone
ETFE	Ethylene tetrafluoroethylene
FE	Friction electrode
CCE	Charge collecting electrode
TL	Triboelectric layer
BD	Breakdown
IL	Insulation layer
EIFE-DC-TENG	DC-TENG with end-isolated FE
SFE-DC-TENG	DC-TENG with split FEs
SCD	Surface charge density
EFI	Electric field intensity
FEM	Finite element method

6) The text has few typos.

Response:

Thank the reviewer for carefully reviewing our research work. We have revised typos and carefully proofread the whole paper, and thank your valuable comments on our research work again.

Reviewer #3 (Remarks to the Author):

The manuscript has introduced cascaded-capacitor breakdown model for DC-TENGs. Considering current understanding of DC-TENG is mostly focused on breakdown mechanism on each electrode, the proposed mechanism seems promising and the experimental results match with simulation data and theoretical analysis. However, there are some comments to be addressed before I can recommend this paper to be published. Detailed comments are as below:

Response:

We highly appreciate the reviewer for the positive comments on our research work.

1. Considering the manuscript is mainly focused on analyzing the working mechanism of DC-TENG, it should include description and appropriate reference of paschen's law and field emission during breakdown process. I believe that explanation on paschen's law and field emission would give in-depth explanation why the electric breakdown occurs as vertical distance and horizontal distance of DC-TENG changes.

Response:

Thank you very much for your careful review and professional comment.

We are very sorry for the lacking of an elaborate explanation about the determination of the electric field intensity threshold required for air breakdown. Here, we provided supplementary information on the detailed analysis process.

In our manuscript, the working condition was in the air of 1 atmosphere, so we calculated the breakdown voltage based on Paschen's law and plotted it in **Fig. R8**. The equation is shown in **Equation R1**, and the values of the parameters in this equation are presented in **Table R2**.

$$V_b = \frac{Bpd}{\ln(Apd) - \ln[\ln(1 + \frac{1}{\gamma})]} \quad (\text{R1})$$

Table R2 Paschen's constants for air

A	B	γ
($\text{m}^{-1} \text{Pa}^{-1}$)	($\text{V m}^{-1} \text{Pa}^{-1}$)	($\times 10^{-3}$)
10.95	273.8	8.136

Figure R8 Paschen's curve: Gas is air at a pressure of 1 atmosphere.

While researchers generally accept the validity of the Paschen's law for distances above 5-10 microns (i.e., the right side of the curve in **Fig. R8**), there is ongoing debate regarding its applicability for smaller air gaps (i.e., the left side of the curve in **Fig. R8**)¹³⁻¹⁵, recently. In our work, we have to utilize the breakdown voltage at distances ranging from ~ 0 (the contact point of the friction electrode (FE) and the triboelectric layer (TL)) to millimeter scale (the distance between FE and charge collecting electrode (CCE)), which falls within the controversial region where the accuracy of Paschen's law is debated. We derived the breakdown threshold of the electric field intensity from the calculated breakdown voltage, which varied from 3 MV m^{-1} to 7 MV m^{-1} for the right side of Paschen's curve. The DC-TENG device itself presents many non-ideal factors that further complicate the analysis of its electrical properties. For instance, the electric field around the DC-TENG is not uniform as assumed in Paschen's law, which is probably due to the mechanical configuration, materials selections, and electromagnetic environment issues. To simplify our model and emphasize the main

focus of this work, we used a breakdown voltage magnitude that many researchers have employed, which is 3 MV m^{-1} ^{14, 16-18}.

For the left side of the curve, some works stated that in addition to the Townsend avalanche of gaseous ions, other effects such as field emission of electrons and tunneling electrons have been proposed as potential factors, and they offered a modified Paschen's curve with a slope of $60\text{-}80 \text{ MV m}^{-1}$ ¹³⁻¹⁵. Whether considering the original Paschen's curve or the modified one, the breakdown electric field intensity exceeds 3 MV m^{-1} by more than an order of magnitude, which does not align with real-world conditions. For instance, the electric field intensity threshold for triggering the field emission is $\sim 75 \text{ MV m}^{-1}$ ¹⁴, requiring ultra-high surface charge density of $\sim 1330 \mu\text{C m}^{-2}$, which is a strict condition to achieve in the air with 1 atmosphere. Hence, 3 MV m^{-1} can be a proper magnitude of breakdown threshold.

During the operation of DC-TENG, the charged TL first enters the 2nd domain, and the 2nd breakdown occurs. As a result, the surface charge density decreases, and then it moves to the 1st domain for the 1st breakdown, which allows the charge to flow through the external circuit. Regardless of whether field emission occurs in the second domain or whether it dominates, the larger the horizontal distance between FE and CCE, the more thorough the 2nd breakdown occurs. This results in more surface charge participating in the 2nd breakdown, leading to less charge participating in the 1st breakdown and subsequently less charge flowing through the external circuit. Regarding the vertical distance between CCE and TL, according to Paschen's law, on the scale of tens to hundreds of micrometers, the larger the vertical distance, the higher the voltage required for air breakdown. This implies that it becomes harder for the 1st breakdown to occur as the vertical distance increases, leading to fewer charges flowing through the external circuit. Thus, more surface charge is left to participate in the 2nd breakdown in the next working cycle.

We are sorry for not introducing Paschen's law in details. To avoid possible confusion, we have added the discussion about Paschen's law in the present supplementary materials and referred to it in the present manuscript.

The revised part in the present manuscript is as follows:

We have made a revision of “As charges continuously flow into the capacitor $C_{\text{CCE-TL}}$, the electric field intensity between CCE and TL keeps increasing until the breakdown threshold, which is 3 MV m^{-1} in the air, is reached. (Discussion about the breakdown threshold is shown in **Supplementary Fig. 1** and **Supplementary Note 3**) When the electric field intensity exceeds the threshold, ...” in the second paragraph of “**Cascaded-capacitor-breakdown model and experimental validation under ideal conditions**”.

The revised part in the present supplementary materials is as follows:

We have added a supplementary figure as follows:

Supplementary Fig. 1 Paschen’s curve: Gas is air at a pressure of 1 atmosphere.

We have added a supplementary note as follows:

“Supplementary Note 3 Determination of the threshold of the electric field intensity base on Paschen’s law

Paschen’s law for gas breakdown gives an empirical relation between the breakdown voltage and the product of the gas pressure and the distance between the

two electrodes involved in the breakdown process. The question is shown in **Equation S1**.

$$V_b = \frac{Bpd}{\ln(Apd) - \ln[\ln(1 + \frac{1}{\gamma})]} \quad (\text{S1})$$

where V_b is the breakdown voltage, p is the gas pressure, d is the distance, A and B are constants determined by the pressure and composition of the gas, and γ is the secondary electron emission coefficient.

In our manuscript, the working condition was in the air of 1 atmosphere, so we plotted the Paschen's curve in **Supplementary Fig. 1** with $A = 10.95 \text{ m}^{-1} \text{ Pa}^{-1}$, $B = 273.8 \text{ V m}^{-1} \text{ Pa}^{-1}$, and $\gamma = 8.136 \times 10^{-3}$.

While researchers generally accept the validity of the Paschen's law for distances above 5-10 microns (i.e., the right side of the curve in **Supplementary Fig. 1**), there is ongoing debate regarding its applicability for smaller air gaps (i.e., the left side of the curve in **Supplementary Fig. 1**)⁴⁻⁶, recently. In our work, we have to utilize the breakdown voltage at distances ranging from ~ 0 (the contact point of the friction electrode (FE) and the triboelectric layer (TL)) to millimeter scale (the distance between FE and charge collecting electrode (CCE)), which falls within the controversial region where the accuracy of Paschen's law is debated. We derived the breakdown threshold of the electric field intensity from the calculated breakdown voltage, which varied from 3 MV m^{-1} to 7 MV m^{-1} for the right side of Paschen's curve. The DC-TENG device itself presents many non-ideal factors that further complicate the analysis of its electrical properties. For instance, the electric field around the DC-TENG is not uniform as assumed in Paschen's law, which is probably due to the mechanical configuration, materials selections, and electromagnetic environment issues. To simplify our model and emphasize the main focus of this work, we used a breakdown voltage magnitude that many researchers have employed, which is 3 MV m^{-1} ^{5, 7-9}.

For the left side of the curve, some works stated that in addition to the Townsend avalanche of gaseous ions, other effects such as field emission of electrons and tunneling electrons have been proposed as potential factors, and they offered a modified Paschen's curve with a slope of $60\text{-}80 \text{ MV m}^{-1}$ ⁴⁻⁶. Whether considering the original

Paschen's curve or the modified one, the breakdown electric field intensity exceeds 3 MV m^{-1} by more than an order of magnitude, which does not align with real-world conditions. For instance, the electric field intensity threshold for triggering the field emission is $\sim 75 \text{ MV m}^{-1}$ ⁵, requiring ultra-high surface charge density of $\sim 1330 \text{ } \mu\text{C m}^{-2}$, which is a strict condition to achieve in the air with 1 atmosphere. Hence, 3 MV m^{-1} can be a proper magnitude of breakdown threshold.”

We have added “

4. D. B. Go & D. A. Pohlman A mathematical model of the modified Paschen's curve for breakdown in microscale gaps. *J. Appl. Phys.* **107**, 103303 (2010).
5. A. J. Wallash & L. Levit Electrical breakdown and ESD phenomena for devices with nanometer-to-micron gaps. *Proc. SPIE* **4980**, 87-96 (2003).
6. A. Peschot, N. Bonifaci, O. Lesaint, C. Valadares & C. Poulain Deviations from the Paschen's law at short gap distances from 100 nm to 10 μm in air and nitrogen. *Appl. Phys. Lett.* **105**, 123109 (2014).
7. D. Liu, et al. A constant current triboelectric nanogenerator arising from electrostatic breakdown. *Sci. Adv.* **5**, eaav6437 (2019).
8. D. Liu, L. Zhou, Z. L. Wang & J. Wang Triboelectric nanogenerator: From alternating current to direct current. *iScience* **24**, 102018 (2021).
9. Z. Zhao, D. Liu, Y. Li, Z. L. Wang & J. Wang Direct-current triboelectric nanogenerator based on electrostatic breakdown effect. *Nano Energy* 107745 (2022).” in the **Reference section of the supplementary materials.**

2. In Figure 5a, the authors have ignored the influence of contact electrification between kapton tape and PVC film. However, since the electrons from the PVC surface constantly flow to electrodes of 1st and 2nd domain, there seems to be constant loss of electrons on PVC surface and would lead to contact electrification of kapton tape and PVC during operation. In addition, as kapton tape is dielectric material, there would be dielectric loss when surface charge of kapton film influences charge of FE through electrostatic induction. This could lead to electrical potential difference between two electrodes of FE part separated with kapton film. The authors should show transferred charge data between two electrodes of FE without the CCE connected to ensure that there is no electron flow (electrical potential difference) between them.

Response:

Thank you very much for your careful review and professional comment. Here we

explained the contact electrification between Kapton type and PVC film and the function of the DC-TENG with split FEs configuration shown in **Fig. 5a** in details.

The contact electrification between Kapton type and PVC film

We agree with you that the Kapton tape will contact triboelectric layer (TL), which is PVC film in our experiments. Although there is contact electrification effect between the Kapton tape and the PVC film, the discharge between them rarely occurs, since the conductive electrode on the other side of the Kapton tape serves as a back electrode to bound charges². In addition, since Kapton is also a kind of dielectric material, after a brief rubbing, the charge on the Kapton tape becomes saturated, and the Kapton tape no longer participates in the contact electrification process, simply acting as an isolating tool with ignorable influence on the electric field or electric output. This part can also be found in **Supplementary Note 17**.

An experiment has been conducted to verify the statement above. Since copper and Kapton have different charge affinity and triboelectrification performance when rubbed against PVC film, we completely covered the surface of the friction electrode (FE) that faces TL and the edge of FE to investigate the role of Kapton in the triboelectrification process of DC-TENG operation. 30 trials were conducted and the charge density collected in each working cycle was calculated, as shown in **Fig. R9**. The data presented in **Fig. 5d** was used for comparison. Compared with the charge density measured in the 1st domain and 2nd domain of DC-TENG with uncovered FE ($\sim 17 \mu\text{C m}^{-2}$ for the 1st domain while $\sim 67 \mu\text{C m}^{-2}$ for the 2nd domain), the charges collected by the DC-TENG with covered FE (configuration is shown in the insert of **Fig. R9**) can be negligible ($\sim 1.3 \mu\text{C m}^{-2}$).

Figure R9 Triboelectric charge densities collected by the DC-TENG with different configurations.

To avoid possible confusion, we have added detailed experimental results and comprehensive descriptions in the supplementary materials.

The revised part in the present manuscript is as follows:

“The isolation layer has little influence on contact electrification and output signals, as discussed in **Supplementary Fig. 13** and **Supplementary Note 17.**” in the fourth paragraph of **“Identification and regulation of three discharge domains”**.

The revised part in the present supplementary materials is as follows:

We have added a supplementary figure as follows:

Supplementary Fig. 13 Triboelectric charge densities collected by the DC-TENG with FE fully covered configuration. Ten working cycles were measured for each trail. The error bar represents the standard deviation.

We have added “As shown in **Supplementary Fig. 13**, compared with the charge density measured in the external circuit of DC-TENG with uncovered FE (dozens of micro coulombs per square meter), that of the DC-TENG with covered FE configuration (insert of **Supplementary Fig. 13**) can be negligible ($\sim 1.3 \mu\text{C m}^{-2}$).” At the end of “**Supplementary Note 17 Influence of the isolated layer on contact electrification process and output signals**”.

The function of the DC-TENG with split FEs shown in Fig. 5a.

Without charge collecting electrode (CCE), there are charges transferred between the two electrodes of FE, and the measured results are shown in **Fig. R10**.

Figure R10 Charge density transferred between the two electrodes of FE on the DC-TENG configuration without CCE.

The purpose of the DC-TENG configuration shown in the **Fig. R11 (Fig. 5a)** is to quantify the charge flow through the 1st and 2nd domains. By isolating the two sections of the conventional FE with Kapton type, the physical effects on the FE are divided into two separate parts. The head electrode of FE (the left section of FE in the diagram) is only responsible for triboelectrification, while the tail one (the right one in the diagram) is only responsible for the 2nd breakdown. From another view, this configuration can be seen as the triboelectrification happens on the head electrode of FE supply charges for two CCEs, the tail electrode of FE (collecting charges participating in the 2nd domain) and the original CCE (collecting charges participating in the 1st domain).

Figure R11 (Fig. 5a) Working mechanism of the DC-TENG with split FEs for quantifying the charges in the 1st and 2nd domains.

Without the original CCE, the tail FE can be considered the sole charge collecting electrode in this configuration, and all charges released by air breakdown will be collected by it. Hence, the charges collected in the DC-TENG without original CCE and with split FEs are larger than the charges collected in the 2nd domain in the original manuscript, as observed in the experimental results.

3. The authors should show matching resistance of the DC-TENG since it is important parameter for application of TENGs. Figure 6d only shows the output power of EIFE-DC-TENG increasing depending on external resistance.

Response:

Thank you very much for your careful review and professional comment.

The matching resistance of DC-TENG has not been clearly defined yet. Ideally, the DC-TENG acts as a constant current source and therefore has no matching resistance, which means the output power keeps increasing with the increase of the external resistance. However, practical issues such as the generalized 2nd and 3rd breakdown, identified in this manuscript, can prevent it from the ideal condition. Correspondingly, the two critical resistances can be considered potential candidates for the matching resistance of the DC-TENG. One is the critical point in the resistance-power plot where the output power no longer increases as the external resistance increases. The other is when the external resistance is sufficiently high to cause the 3rd breakdown between the two electrodes (FE and CCE).

For the conventional DC-TENG employed in **Fig. 6d**, further external load experiments (**Fig. R12**) demonstrate that the output power remains constant at a certain magnitude of 4-8 GΩ. A possible explanation is that a large resistor limits the charge

transfer rate in the external circuit, leading to an accumulation of charges at the FE, and consequently, some of the accumulated charges are released from the 2nd domain. Whether the 3rd breakdown occurs depends mainly on the relative magnitude between the rate of triboelectric charge generation and the total rate of charge released from the 1st and 2nd domains. Based on the experimental results of the conventional DC-TENG with a 2 mm distance between FE and CCE, the 3rd breakdown does not occur until an external resistance of ~ 20 G Ω .

Figure R12 Output power of the conventional DC-TENG under different external resistance.

4. In addition, considering conventional TENG has matching resistance around 10 to few hundred megaohm, the EIFE-DC-TENG seems to have quite high matching resistance over 3 gigaohm. Can authors explain the reason why it shows such high matching resistance?

Response:

Thank you very much for your careful review and professional comment.

Unlike the matching resistance of the conventional TENG³⁷, that of the DC-TENG has not been clearly defined yet. For the conventional TENG based on electrostatic induction, the matching resistance can be calculated by the governing equation derived from a capacitor model. The matching resistance, namely the optimized resistance, is related to the motion characteristics, which influence the varying capacitance of the TENG device³⁷. Ideally, DC-TENG can be considered as a constant current source with

fixed inherent capacitance (fixed mechanical configuration), which does not have matching resistance and the output power keeps increasing with the increase of the external resistance. Nevertheless, the generalized 2nd and 3rd breakdowns presented in this manuscript prevent the DC-TENG from an ideal constant current source. Hence, the two critical resistances can be considered potential candidates for the matching resistance of the DC-TENG. One is the critical point in the resistance-power plot where the output power no longer increases as the external resistance increases. The other is when the external resistance is sufficiently high to cause the 3rd breakdown between the two electrodes (FE and CCE).

Figure R13 Output power under different external load conditions. a with, and **b** without the 2nd domain blocked (the insert table shows the critical resistance where the 3rd breakdown occurs under different motion frequencies). The output current signals under the 3rd breakdown resistance of **c** 2.828 Hz **d** 2.000 Hz **e** 1.414 Hz motion frequencies.

For the conventional DC-TENG, more external load experiments have been conducted (**Fig. R13a**), which revealed that the output power stopped increasing beyond an external load of $\sim 4 \text{ G}\Omega$. Besides, the 3rd breakdown did not occur until $20 \text{ G}\Omega$, suggesting that more charges were released from the 2nd domain instead of

remaining on the friction electrode (FE), preventing the potential difference between the two electrodes from reaching the breakdown threshold. Moreover, although it is expected that the current would increase with increasing frequencies, **Fig. R13a** reveals only a marginal increase in current as the motion frequency increased from 1.414 Hz to 2.000 Hz, while virtually no difference between the current at motion frequencies of 2.000 Hz and 2.828 Hz. This observation may be because a larger rate of charge generation leads to a larger rate of charge release in the 2nd domain. The above-discussed phenomena further indicate that the 2nd breakdown cannot be ignored.

The DC-TENG with end-isolated FE, as proposed in this manuscript, effectively blocked the 2nd breakdown by filling the 2nd domain with Kapton. This modification makes the practical DC-TENG approach ideal behavior. Consequently, the 3rd breakdown is the only limiting factor for further increasing the output power. With the configuration illustrated in **Fig. 6d** insert, more experiments have been conducted to further characterize the output performance (**Fig. R13b**). As expected, higher frequency (higher sliding speed) leads to obviously higher output power. As the 2nd domain is blocked, the triboelectric charges accumulated on the FE due to the limitation of high resistance in the external circuit, raising the potential difference between FE and charge collecting electrode (CCE), resulting in the 3rd breakdown. As **Fig. R13c, d, and e** show, with higher motion frequencies, the current and the corresponding voltage are higher. Hence, higher sliding speed leads to a lower critical resistance for the 3rd breakdown, which supported that the ideal DC-TENG is a constant current source, but the existence of the 2nd and 3rd breakdowns limits the further increase of the output power as the external resistance increases. In summary, the matching resistance of conventional TENG is due to the varying capacitance and can be calculated by the governing formula derived from the capacitor model, while the ideal DC-TENG can be considered as a constant current source that does not have a matching resistance. The two critical values of resistance shown above are due to the non-ideal issues, which can be partially solved by isolating the end of FE.

References in Response:

1. Z. Zhao, et al. Rationally patterned electrode of direct-current triboelectric nanogenerators for ultrahigh effective surface charge density. *Nat. Commun.* **11**, 1-9 (2020).
2. W. He, et al. Boosting output performance of sliding mode triboelectric nanogenerator by charge space-accumulation effect. *Nat. Commun.* **11**, 1-8 (2020).
3. M. Siek, W. Adamkiewicz, Y. I. Sobolev & B. A. Grzybowski The influence of distant substrates on the outcome of contact electrification. *Angew. Chem. Int. Ed.* **57**, 15379-15383 (2018).
4. Y. I. Sobolev, W. Adamkiewicz, M. Siek & B. A. Grzybowski Charge mosaics on contact-electrified dielectrics result from polarity-inverting discharges. *Nat. Phys.* **18**, 1347-1355 (2022).
5. J. Lowell & A. Rose-Innes Contact electrification. *Adv. Phys.* **29**, 947-1023 (1980).
6. Y. Liu, et al. Quantifying contact status and the air-breakdown model of charge-excitation triboelectric nanogenerators to maximize charge density. *Nat. Commun.* **11**, 1599 (2020).
7. J.-S. Chang, P. A. Lawless & T. Yamamoto Corona discharge processes. *IEEE Trans. Plasma Sci.* **19**, 1152-1166 (1991).
8. K. Nakayama & R. A. Nevshupa Plasma generation in a gap around a sliding contact. *J. Phys. D: Appl. Phys.* **35**, L53 (2002).
9. P. Thiessen Physical and chemical investigations in tribomechanical events. *Z Chem* **5**, 162-71 (1965).
10. K. Nakayama & H. Hashimoto Triboemission from various materials in atmosphere. *Wear* **147**, 335-343 (1991).
11. K. Nakayama, N. Suzuki & H. Hashimoto Triboemission of charged particles and photons from solid surfaces during frictional damage. *J. Phys. D: Appl. Phys.* **25**, 303 (1992).
12. K. Nakayama & F. Yagasaki The Flow of Triboplasma. *Tribol. Lett.* **67**, 72 (2019).
13. D. B. Go & D. A. Pohlman A mathematical model of the modified Paschen's curve for breakdown in microscale gaps. *J. Appl. Phys.* **107**, 103303 (2010).
14. A. J. Wallash & L. Levit Electrical breakdown and ESD phenomena for devices with nanometer-to-micron gaps. *Proc. SPIE* **4980**, 87-96 (2003).
15. A. Peschot, N. Bonifaci, O. Lesaint, C. Valadares & C. Poulain Deviations from the Paschen's law at short gap distances from 100 nm to 10 μm in air and nitrogen. *Appl. Phys. Lett.* **105**, 123109 (2014).
16. D. Liu, et al. A constant current triboelectric nanogenerator arising from electrostatic breakdown. *Sci. Adv.* **5**, eaav6437 (2019).
17. D. Liu, L. Zhou, Z. L. Wang & J. Wang Triboelectric nanogenerator: From alternating current to direct current. *iScience* **24**, 102018 (2021).
18. Z. Zhao, D. Liu, Y. Li, Z. L. Wang & J. Wang Direct-current triboelectric nanogenerator based on electrostatic breakdown effect. *Nano Energy* 107745 (2022).
19. V. Nguyen & R. Yang Effect of humidity and pressure on the triboelectric nanogenerator. *Nano Energy* **2**, 604-608 (2013).
20. X.-S. Zhang, et al. High-performance triboelectric nanogenerator with enhanced energy density based on single-step fluorocarbon plasma treatment. *Nano Energy* **4**, 123-131 (2014).
21. V. Nguyen, R. Zhu & R. Yang Environmental effects on nanogenerators. *Nano Energy* **14**, 49-61 (2015).

22. R. Wen, J. Guo, A. Yu, J. Zhai & Z. I. Wang Humidity-resistive triboelectric nanogenerator fabricated using metal organic framework composite. *Adv. Funct. Mater.* **29**, 1807655 (2019).
23. Y. Hu, X. Wang, H. Li, H. Li & Z. Li Effect of humidity on tribological properties and electrification performance of sliding-mode triboelectric nanogenerator. *Nano Energy* **71**, 104640 (2020).
24. L. Li, et al. The electron transfer mechanism between metal and amorphous polymers in humidity environment for triboelectric nanogenerator. *Nano Energy* **70**, 104476 (2020).
25. K. Wang, et al. Effect of relative humidity on the enhancement of the triboelectrification efficiency utilizing water bridges between triboelectric materials. *Nano Energy* **93**, 106880 (2022).
26. Z. Li, B. Xu, J. Han, J. Huang & K. Y. Chung Interfacial polarization and dual charge transfer induced high permittivity of carbon dots-based composite as humidity-resistant tribomaterial for efficient biomechanical energy harvesting. *Adv. Energy Mater.* **11**, 2101294 (2021).
27. H. Zhang, et al. Triboelectric nanogenerator as self-powered active sensors for detecting liquid/gaseous water/ethanol. *Nano Energy* **2**, 693-701 (2013).
28. K. Y. Lee, et al. Hydrophobic sponge structure-based triboelectric nanogenerator. *Adv. Mater.* **26**, 5037-5042 (2014).
29. V.-T. Bui, J.-H. Oh, J.-N. Kim, Q. Zhou & I.-K. Oh Nest-inspired nanosponge-Cu woven mesh hybrid for ultrastable and high-power triboelectric nanogenerator. *Nano Energy* **71**, 104561 (2020).
30. W. Du, et al. Inflammation-free and gas-permeable on-skin triboelectric nanogenerator using soluble nanofibers. *Nano Energy* **51**, 260-269 (2018).
31. D. Liu, et al. Performance enhanced triboelectric nanogenerator by taking advantage of water in humid environments. *Nano Energy* **88**, 106303 (2021).
32. N. Wang, et al. New hydrogen bonding enhanced polyvinyl alcohol based self-charged medical mask with superior charge retention and moisture resistance performances. *Adv. Funct. Mater.* **31**, 2009172 (2021).
33. N. Wang, Y. Zheng, Y. Feng, F. Zhou & D. Wang Biofilm material based triboelectric nanogenerator with high output performance in 95% humidity environment. *Nano Energy* **77**, 105088 (2020).
34. R. Zhang, et al. Cellulose-based fully green triboelectric nanogenerators with output power density of 300 W m⁻². *Adv. Mater.* **32**, 2002824 (2020).
35. L. Liu, et al. A high humidity-resistive triboelectric nanogenerator via coupling of dielectric material selection and surface-charge engineering. *J. Mater. Chem. A* **9**, 21357-21365 (2021).
36. L. Liu, et al. Achieving ultrahigh effective surface charge density of direct-current triboelectric nanogenerator in high humidity. *Small* **18**, 2201402 (2022).
37. S. Niu, et al. Theoretical study of contact-mode triboelectric nanogenerators as an effective power source. *Energy Environ. Sci.* **6**, 3576-3583 (2013).

REVIEWERS' COMMENTS

Reviewer #1 (Remarks to the Author):

The revised manuscript has addressed my question, and is possible to be accepted for publication.

Reviewer #2 (Remarks to the Author):

The authors have done an excellent job addressing all the concerns raised by the reviewers and clarifying several other important issues. It is evident to me that there is triboplasma (corona discharge) present under low humidity conditions. Therefore, I recommend that the paper be published.

Reviewer #3 (Remarks to the Author):

The authors addressed every comment carefully from the last review.

Point-by-point responses to the reviewers' comments

We sincerely thank the reviewers for carefully reviewing our work, which are indeed very helpful to make the paper more solid and smooth. We have revised our manuscript in accordance with your pertinent comments and suggestions. The following responses are prepared to address all of the reviewers' comments in a point-by-point fashion. (**Comments in Black, responses in Blue.**)

REVIEWER COMMENTS

Reviewer #1 (Remarks to the Author):

The revised manuscript has addressed my question, and is possible to be accepted for publication.

Response:

We highly appreciate the reviewer for carefully reviewing our work, and thank your valuable comments on our research work and recommendation for publications.

Reviewer #2 (Remarks to the Author):

The authors have done an excellent job addressing all the concerns raised by the reviewers and clarifying several other important issues. It is evident to me that there is triboplasma (corona discharge) present under low humidity conditions. Therefore, I recommend that the paper be published.

Response:

We highly appreciate the reviewer for carefully reviewing our work, and thank your contribution to our research work and recommendation for publications.

Reviewer #3 (Remarks to the Author):

The authors addressed every comment carefully from the last review.

Response:

We highly appreciate the reviewer for carefully reviewing our work, and thank your positive comments and valuable feedback on our research work.